

# Time varying changes in the simulated structure of the Brewer Dobson Circulation

Chaim I Garfinkel[1], Valentina Aquila[2], Darryn W Waugh[2], and Luke D Oman[3]

[1]The Fredy and Nadine Herrmann Institute of Earth Sciences, Hebrew University, Jerusalem, Israel.
[2] Department of Earth and Planetary Science, Johns Hopkins University, Baltimore, MD, USA
[3] NASA Goddard Space Flight Center, Greenbelt, MD, USA.

*Correspondence to:* Chaim I. Garfinkel (chaim.garfinkel@mail.huji.ac.il)

**Abstract.** A series of simulations using the NASA Goddard Earth Observing System Chemistry-Climate Model are analyzed in order to assess changes in the Brewer-Dobson Circulation (BDC) over the past 55 years. When trends are computed over the past 55 years, the BDC accelerates throughout the stratosphere, consistent with previous modeling results. However, over the second half of the simulations (i.e. since the late 1980s), the model simulates structural changes in the BDC as the tem-

5 poral evolution of the BDC varies between regions in the stratosphere. In the mid-stratosphere in the mid-latitude Northern Hemisphere, the BDC decelerates in a simulation despite increases in greenhouse gas concentrations and warming sea surface temperatures. This deceleration is reminiscent of changes inferred from satellite instruments and in-situ measurements. In contrast, the BDC in the lower-stratosphere continues to accelerate. The main forcing agents for the recent slowdown in the mid-stratosphere appear to be declining ODS concentrations and the timing of volcanic eruptions. Changes in both age of air

and the tropical upwelling of the residual circulation are similar. We therefore clarify that the statement that is often made that climate models simulate a decreasing age throughout the stratosphere only applies over long time periods, and is not the case for the past 25 years when we have most tracer measurements.

## 1 Introduction

The global circulation in the stratosphere - the Brewer-Dobson circulation (BDC) - consists of air-masses rising across the

15 tropical tropopause, moving poleward, and sinking into the extratropical troposphere (Holton et al., 1995; Waugh and Hall, 2002; Butchart, 2014). Since the BDC and its changes have important implications on both stratospheric and tropospheric climate as well as stratospheric ozone chemistry (SPARC-CCMVal, 2010; World Meteorological Organization, 2011, 2014; Manzini et al., 2014), it is important to assess the factors that lead to simulated BDC changes and whether historical changes in the BDC as simulated by models are consistent with available observational constraints.

The BDC has historically been deduced either from the residual circulation or from the average time for an air parcel to travel from the tropical troposphere to a given stratospheric sampling region (i.e. the mean age of air or mean age). While these two diagnostics are clearly related, differences can arise due to isentropic mixing and recirculation (Waugh and Hall, 2002; Strahan et al., 2009; Li et al., 2012). Specifically, the vertical component of the residual circulation ($\overline{w*}$) measures the instantaneous advection whereas mean age is an integrated measure of the total transport. Only in the tropical lower stratosphere,





where the age is young and the flow can be thought of as dominated by vertical advection, is there a reason to expect the two metrics to indicate a similar evolution.

Chemistry climate models robustly predict a strengthened BDC under climate change in the middle and lower stratosphere of approximately $1-5\%$ per decade (the precise rate depends on the level considered and varies among models;
Butchart and Scaife, 2001; Butchart et al., 2006; Garcia and Randel, 2008; Li et al., 2008; Waugh, 2009; Shepherd and McLandress, 2011; Garcia et al., 2011; Lin and Fu, 2013; Butchart, 2014; Oberländer-Hayn et al., 2015). The model used in this study, the Goddard Earth Observing System Chemistry-Climate Model, Version 2 (GEOSCCM), predicts a trend quantitatively similar to those in other models both for the historical period and for the future (Oman et al., 2009; Waugh, 2009; Butchart et al., 2010; Li et al., 2012).

It has been argued that observational estimates of historical changes do not agree with the simulated acceleration trend. Specifically, the analysis of historical tracer data does not provide evidence for an acceleration trend in the mid-stratosphere Northern Hemisphere (NH), where mean age actually appears to have increased (Engel et al., 2009; Bönisch et al., 2011; Stiller et al., 2012; Hegglin et al., 2014; Ray et al., 2014). In particular, the mean age evolution in the figures of Engel et al. (2009) and Ray et al. (2014) indicates pronounced aging since the late 1980s, with earlier changes less clear. Reanalysis data
also suggests aging of NH mid-stratosphere air (Diallo et al., 2012; Monge-Sanz et al., 2013; Ploeger et al., 2015) since the late 1980s. While these observational and reanalysis based studies disagree about the sign of changes in other regions of the stratosphere, they all indicate aging of the NH midlatitude mid-stratosphere. Ray et al. (2014) argue that the large uncertainty estimates on the trends presented by Engel et al. (2009) are overly conservative, and that this aging trend is statistically significant. Several of the aforementioned studies suggest that these changes in the mean age imply a redistribution of the BDC, and
specifically a slowdown of the deep (i.e. mid-stratospheric) NH branch of the BDC and/or less mixing of fresh tropical air into this region. However, it is not clear what forcings (if any) could be responsible for this redistribution and also whether models of the kind used in WMO-Ozone Assessments (World Meteorological Organization, 2011, 2014) can capture such a slowdown given known forcings.

Oman et al. (2009) conducted time-slice and transient simulations using a previous version of the model used in this study,
and they found a trend towards younger mean age everywhere in the stratosphere between 1960 and 2004, though the decrease in mean age in the Southern Hemisphere (SH) is larger due to Antarctic ozone depletion. Oman et al. (2009) also found that ozone recovery would lead to a slowdown of the BDC if not for warming SSTs due to increasing greenhouse gas emissions. Oberländer-Hayn et al. (2015) recently presented differences in mean age and the residual circulation in time-slice simulations using the ECHAM/MESSy Atmospheric Chemistry Model. The changes in tropical upward mass flux indicate a strengthening
of the BDC between 1960 and 2000 in the NH winter season in the lower and a weakening in the upper stratosphere with a change in sign at 10hPa. Changes in mean age show a decrease of about 0.13yr/decade in the lower and middle stratosphere and a slight increase in the Arctic upper stratosphere and lower mesosphere. While there is some hint of a structural change in the properties of the BDC, the changes occur higher in the stratosphere and at more polar latitudes than is suggested by available observations.





The goal of this study is to understand whether historical forcings have led to structural changes in the BDC. As it is difficult to compare historical tracer observations to time-slice simulations, we focus on transient simulations. In order to understand the forcings that may have led to the structural changes, we start with an ensemble in which the only time-varying forcing is changing SSTs and sea ice, and then sequentially add the following forcings: greenhouse gases, ozone depleting substances, volcanic eruptions, and solar variability. Note that neither Oberländer-Hayn et al. (2015) nor Oman et al. (2009) considered volcanic forcings and solar variability, and we want to understand their possible influence. In order to (partially) damp internal atmospheric variability, three ensemble members are performed for each experimental setup. Finally, the simulations extend from January 1960 through December 2014 in order to capture the full historical record.

We show that over the full duration of the experiments (i.e for a start-date in 1960), we recover the result from previous modeling studies: anthropogenic climate change leads to acceleration of the BDC throughout the stratosphere. However, our model can simulate statistically significant aging of the mid-latitude NH near 20hPa between the early 1990s and the present, and thus is qualitatively consistent with available observations. This suggests that structural changes of the BDC did occur since the late 1980s: the BDC accelerated in the lower stratosphere, but decelerated in the mid-stratosphere, in both the tropics and in the NH. Mean age and the residual circulation (as measured by tropical $\overline{w*}$) change in unison. The cause of this deceleration trend is a combination of forcings - ODS recovery and the timing of volcanic eruptions - that together outweighed greenhouse gas induced acceleration since the late 1980s. We therefore emphasize that if one wishes to capture observed historical changes, careful attention must be paid to the start and end dates used for trend calculation and the forcings included in a model simulation.

## 2 Methods

GEOSCCM is an aerosol and chemistry focused version of the GEOS-5 Earth system model, including radiatively and chemically coupled tropospheric and stratospheric aerosol and atmospheric chemistry. GEOSCCM couples the GEOS-5 (Rienecker et al, 2008; Molod et al., 2012) atmospheric general circulation model to the comprehensive stratospheric chemistry module StratChem described in Pawson et al. (2008), and the Goddard Chemistry, Aerosol, Radiation, and Transport Model (GOCART) described in Colarco et al. (2010). Previous versions of GEOSCCM have been graded highly in the two phases of the Chemistry-Climate Model Validation (Eyring et al., 2006; SPARC-CCMVal, 2010). Improvements to the model since then are described in Oman and Douglass (2014) and Aquila et al. (2016).

A series of simulations of the period from January 1960 to December 2014 have been performed in order to understand the past evolution of the stratosphere. These simulations were presented in Aquila et al. (2016), where the focus was on changes in temperatures. Here we examine changes in the BDC. These simulations are grouped into the following five experiments:

1. **SST**, which uses time-varying observed sea surface temperatures (SSTs) and sea ice up to November 2006 from the MetOffice Hadley Centre observational dataset (Rayner et al., 2006) and from Reynolds et al. (2002) and updates since then (Figure 1a). GHGs and ODS concentrations are fixed at 1960-levels. Volcanic eruptions are not included in this experiment, and the solar forcing is held constant;



2. **SSTGHG**, which includes observed SSTs and increasing GHG concentrations (Figure 1b). GHG concentrations are from observations up to 2005 and from the Representative Concentrations Pathway 4.5 after 2005 (Meinshausen et al., 2011);

3. **SSTGHGODS**, which includes observed SSTs, increasing GHGs, and changing ODS concentrations following
World Meteorological Organization (2011, Figure 1c);

4. **SSTGHGODSVOL**, which includes observed SSTs, increasing GHGs, changing ODS, and volcanic eruptions, specified after Diehl et al. (2012) from 1979 to December 2010 and Carn et al. (2015) from January 2011 to December 2014 (Figure 1d). The only eruption before 1979 included is Mt. Agung in 1963;

5. **SSTGHGODSVOLSOL (or All-forcing)**, which includes observed SSTs, increasing GHGs, changing ODS, volcanic
eruptions and changes in solar flux as in Lean (2000) and subsequent updates (Figure 1e).

Each experiment is composed of three ensemble members initialized with different initial conditions from a 1960 time-slice simulation. Because we have three members for each forcing combination, we can also assess at least partially the range of internal atmospheric variability. All simulations used emissions of tropospheric aerosol and aerosol precursors following Granier et al. (2011).

The use of observed SSTs in our simulations, rather than internally calculated by the model, produces a climate state closer to the observed one. However, partitioning trends into an SST driven component and a component from other radiative or chemical forcings is somewhat artificial, as the prescribed SST changes occur in response to and in tandem with the changing direct atmospheric forcing; however such a partitioning is an effective tool for disentangling the physical mechanisms leading to changes in the atmospheric circulation (Deser and Phillips, 2009). Specifically, Oman et al. (2009) found that the mean
tropospheric warming associated with rising SSTs had a bigger impact on mean age than the direct radiative impacts of $CO_2$. Note that interannual and decadal variability in SSTs drives changes in the BDC that likely has nothing to do with climate change, and hence we include a smoothed version of the SST variations in Figure 1a. We assume that BDC perturbations induced by each forcing agent adds linearly to the others, as previous work focusing on forcing agents for lower stratospheric mass flux and mid-stratospheric mean age suggests that non-linearities are small (e.g. Oman et al., 2009; McLandress et al.,
2010). The model version used to perform these integrations is no longer supported, and hence we cannot explicitly test this assumption.

A passive tracer is used to derive the mean age. The mixing ratio of this tracer increases linearly with time, and the time lag in tracer concentrations between a certain grid point in the stratosphere and a reference point in the troposphere provides an estimation of mean age at this stratospheric grid point. We adopt a reference point of 200hPa at the equator. All integrations
are branched off from a long time-slice control run with 1960 conditions and so mean age can be defined at the beginning of the experiments. Note that diagnostic output necessary to compute the full age spectrum was not saved for these model experiments, and hence we are limited in our ability to quantify mixing changes.





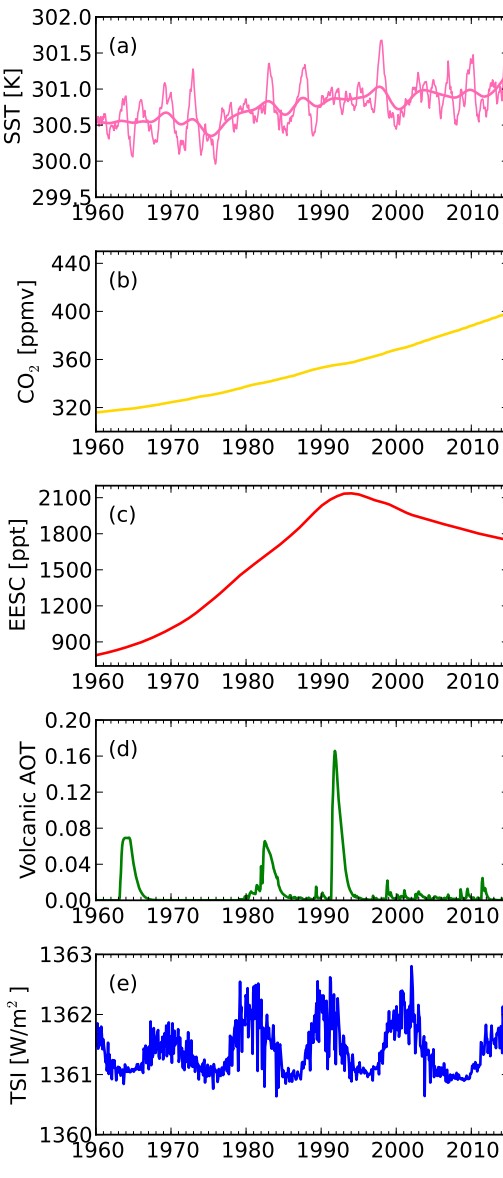

**Figure 1.** Forcing applied in the simulations. a) 10S-10N average sea surface temperatures; b) Atmospheric concentrations of $CO_2$ following RCP4.5; c) equivalent effective stratospheric chlorine (EESC Newman et al., 2007, equation 1 with a = 1 and $\alpha$ for $Br_y = 60$, and using 3-year mean age); d) ensemble mean of the aerosol optical thickness from explosive volcanic eruptions, resulting from prescribed injections of volcanic SO2; e)total solar irradiance.





The aging of NH mid-stratospheric air in observations is pronounced mainly after the late 1980s (see figures in Engel et al., 2009; Ray et al., 2014), and several of the reanalysis-based studies begin their analysis in the late 1980s. Hence, in our presentation and discussion of the results, we consider trends and variability both over the full period of integration and also after the late 1980s. Finally, global mean age profiles as retrieved by satellites are only available since 2002 (Stiller et al., 2012; Haenel et al., 2015), and hence we show trends since 2002 separately as well.

The trends are calculated with a linear least squares fit. Statistical significance of the trends in individual ensemble members of GEOSCCM are computed using a 2-tailed Student's t-test, and the reduction in degrees of freedom due to autocorrelation of the residuals is taken into account with the formula $N(1 - r_1)(1 + r_1)^{-1}$, where n is the number of years and $r_1$ is the lag-1 autocorrelation (eq. 6 of Santer et al., 2008). In computing the ensemble mean response, we first averaged the mean age among the three ensemble members for each season/year (in order to damp the internal, unforced, atmospheric variability), and then compute the trend based on the ensemble mean mean age.

## 3   Results

We now examine how the BDC as simulated by the GEOSCCM has changed in structure over the past 55 years. We show that BDC changes (or "trends") vary with period considered and location. These structural changes are associated with several distinct forcings, and these forcings transiently drive changes in the BDC. When combined with internal variability, it is possible that these forcings can drive, over the 27 year period of 1988-2014, a deceleration trend in the NH mid-stratosphere.

### 3.1   Changes in the All-forcing ensemble

We begin with changes in the BDC for the all forcing ensemble. We first consider changes in the residual circulation and in mean age as a function of latitude and pressure over the full duration of the experiment in the ensemble mean (Figure 2). The BDC accelerates throughout the stratosphere as mean age decreases and the residual circulation accelerates. Hence, changes over the full integration period in our experiments are consistent with previous work (e.g. Oman et al., 2009). The changes are statistically significant at the 95% level from 100hPa and upwards at all latitudes in Figure 2, and all three ensemble members indicate qualitatively similar acceleration trends.

However, the ensemble mean acceleration trend weakens (and even reverses locally) and its robustness goes away as we consider more recent periods. To demonstrate this, we start by showing trends since 1988 in Figure 3. In the lower stratosphere (i.e. the shallow branch of the BDC), the BDC continues to accelerate in all three ensemble members, and this change is statistically significant at the 95% level in each ensemble member individually and in the ensemble mean. In the mid- and upper- stratosphere, however, trends are not robust across the various ensemble members. One of the three ensemble members shows decreasing mean age and an accelerated residual circulation (Figure 3a), while another shows the opposite (Figure 3e). In this ensemble member with aging air, upwelling decreases throughout the tropics, such that both the residual circulation and mean age diagnostics suggest deceleration of the BDC. None of the mean age trends in the mid- and upper- stratosphere in Figure 3ace are statistically significant at the 95% level. Note that if the start-date for the trend is advanced to 1992 then





all three ensemble members indicate deceleration of the BDC in the NH midlatitude mid-stratosphere (shown later), but we prefer to demonstrate that even for a start date of 1988, aging can be simulated given the large amount of internal atmospheric variability.

These differences in trends since 1988 among ensemble members can be reconciled with the changes in the wave forcing of 5 the BDC. Previous modeling and theoretical work has demonstrated that changes in the wave forcing directly force changes in the residual circulation (Rosenlof, 1995; Butchart, 2014). Specifically, enhanced wave convergence (i.e. deceleration of the mean flow) leads to enhanced upwelling on the tropical flank of the enhanced wave convergence and downwelling on the poleward flank (Haynes et al., 1991). As both resolved wave and gravity waves are important for the total wave driving, we evaluate their combined impact in common units of acceleration of the mean flow (m/s/day/decade); positive values indicates 10 less wave convergence and deceleration of the residual circulation, while negative values indicates enhanced wave convergence and acceleration of the residual circulation. The right column of Figure 3 demonstrates that the difference between the ensemble member with a weakened SH residual circulation and the one with an accelerated SH circulation is related to differences in the wave driving. For the ensemble member with a weakened residual circulation in the SH (and thus aging of air in the mid-stratosphere), there is less wave flux converging in the SH ( Figure 3f). In contrast, for the member with an accelerated residual 15 circulation in the SH (and thus decreasing mean age in the NH mid-stratosphere), there is enhanced wave flux converging in the SH ( Figure 3b). Hence, the difference between aging and freshening of mid-stratospheric air is associated with the internal atmospheric variability associated with wave fluxes. Note that in all three ensemble members there is reduced wave convergence in the NH stratosphere, and this effect is most pronounced in winter (not shown). The cause of the decrease in NH stratospheric wave convergence is discussed in Garfinkel et al. (2015) and in the next section. Overall, our model simulations 20 indicate that known forcings could have lead to a slowdown of the deep branch of BDC since 1988 given the large amount of internal variability in wave fluxes.

We now consider the time evolution of changes in mean age in various regions in Figure 4. Changes in the NH mid-latitudes in the mid-stratosphere are considered in Figure 4a. Note that this is the region where available observations suggest that air has aged over the past several decades. In the All-forcing ensemble (blue line), mean age decreases by 0.6 years between 1960 and 25 1992 (i.e. 0.2 years per decade), but then ages by 0.2 years since 1992 (i.e. 0.1 years per decade). The aging trend since 1992 is statistically significant at the 95% level in two of the three ensemble members and in the ensemble mean. Figure 5 considers the evolution of each of the three ensemble members individually. The ensemble spread in mean age in any given year can exceed 5%. As discussed above, one ensemble member simulate aging trends if the start-date of the trend calculation is set at 1988 (this is the ensemble member with anomalously old air after 2010). A different ensemble member simulates anomalously 30 younger air relative to the other three after El Chichon. However, it is conceivable that if these two anomalies occurred in the same integration then aging mean air trends could be simulated with even earlier start dates.

In the tropical and SH mid-stratosphere, mean age is largely unchanged from 1990 through the end of the simulation (blue curves in Figure 4bc). In contrast, in the lower stratosphere, mean age continues to decline in the tropics and in the NH, though not in the SH (blue curves in Figure 4def). The decline in mean age is statistically significant at the 95% level in two of three



ensemble members and in the ensemble mean in the NH lower stratosphere. Hence, there are time- and space-varying variations in recent mean age trends, and only in some regions of the stratosphere has mean age continued to decline.

Similar structural changes are evident for the tropical residual vertical velocity ($\overline{w*}$, Figure 6). In the all-forcing experiment, tropical upwelling accelerated until 1990 in both the mid-stratosphere and lower stratosphere (i.e. blue line rises in both panels of Figure 6), but since 1990 has decreased at all levels above 70hPa (e.g. Figure 6a; (the 70hPa evolution is shown in Polvani et al., 2016,  )). These changes are reminiscent of those proposed by Ray et al. (2010) and Ray et al. (2014) in order to explain how NH midlatitude air in the mid-stratosphere anomalously ages. Specifically, Ray et al. (2010) concluded that for a tropical leaky pipe model, "the best quantitative agreement with the observed mean age and ozone trends over the past three decades is found assuming a small strengthening of the mean circulation in the lower stratosphere, [and] a moderate weakening of the mean circulation in the middle and upper stratosphere", as simulated by GEOSCCM. Overall, it is clear that GEOSCCM can simulate structural changes in the BDC that resemble those inferred from observations.

## 3.2 Forcing of the Trends

We now consider the forcing mechanisms behind these structural changes in the BDC.

### 3.2.1 Mid-Stratosphere

As shown above, mean age in the NH mid-latitudes in the mid-stratosphere decreases by 0.6 years between 1960 and 1992 in the All-forcing ensemble, but then ages by 0.2 years since 1992 (blue line; Figure 4a). This evolution can be broken down into its various forcing components.

1. **SSTs** The red line in Figure 4a shows that SSTs lead to a decrease in mean age of 0.1 years over the course of these 55 years, but with substantial interannual and decadal variability. Specifically, over the last 27 years of the integrations (from 1988 to the end), there is no discernible change. The cause of this is a reduction in planetary wave flux entering the stratosphere. As shown by Garfinkel et al. (2015), recent changes in SSTs have led to a decline in planetary wave flux (especially wave 1) entering the NH stratosphere in midlatitudes: the vertical component of the Eliassen Palm flux at 100hPa area averaged between 40N-80N declines in all three all forcing and in all three SST only experiments, and the decrease is statistically significant at the 95% level in the ensemble mean in both January through March (the focus of Garfinkel et al., 2015) and in the annual mean. This decline in upward propagating midlatitude planetary waves at 100hPa impacts the deep branch more strongly (Plumb, 2002; Ueyama et al., 2013), though it is compensated by the otherwise expected acceleration due to warming oceans (figure 1a and  Oman et al., 2009). Hence, it is not surprising that little change occurred over the last 27 years of the integration.

2. **Greenhouse gases** The influence of GHG gases can be deduced from the difference between the red and magenta curves, as the SST only ensemble is conducted with radiative forcings fixed at 1960 levels. The difference between the curves exceeds 0.1years towards the end of the integrations.



3. **Declining ODS concentrations** The effect of increasing ODS concentrations can be deduced from the difference be-
   tween the magenta and green curves, and it suggests that increasing ODS concentrations led to a decrease in mean age of
   0.3 years by 1995 (Figure 4a). More recently, declining ODS concentrations (Figure 1c) lead to a recovery towards older
   air; that is, the gap between the magenta and green curves decreases between 1995 and the present. Note that Oman et al.
   (2009) also found that ozone recovery leads to a slowdown of the BDC in a previous version of the model we use. Finally,
   we note the caveat that while declining ODS concentrations clearly impact the BDC in these integrations, a statistically
   significant recovery of ozone has been detected in observations only in the upper stratosphere in mid-latitudes and the
   tropics (World Meteorological Organization, 2014).

4. **Volcanos and solar** The influence of volcanic eruptions can be deduced from the difference between the green and
   cyan curves, and in our model simulations the eruption of Mt Pinatubo and El Chichon led to a decrease in mean age
   approaching 0.3 years, which gradually decayed over four to six years. Minor volcanic eruptions in the past ten years
   may have led to an additional decrease in mean age of 0.05 to 0.1 years. The net effect is that large eruptions (or lack
   thereof) can influence decadal variability in mean age in GEOSCCM. Solar influences appear to be relatively minor, and
   we therefore focus our attention on the other forcings in this paper (Figure 4a).

The net effect is that over the second half of the experiments (when observations are more numerous), decreasing ODS con-
centrations and the recovery from Pinatubo overcame the influence of rising GHG concentrations and led to aging of 0.2
years.

In the tropical mid-stratosphere, mean age in the all-forcing ensemble does not change from 1990 through the end of the sim-
ulation (blue curve in Figure 4b). Changes in this region are dominated by ODS concentrations, and since ODS concentrations
recover after the mid-1990s (Figure 1c), mean age does not change despite rising GHG concentrations. ODS concentrations
also dominate SH mid-stratospheric mean age evolution (Figure 4c), and since ODS concentrations decrease after 1995, mean
age is flat since 1990 in the all-forcing ensemble as well.

### 3.2.2 Lower Stratosphere

In the tropical and NH lower stratosphere (Figure 4d-e), increasing GHG concentrations and warming SSTs drive a decrease in
mean age throughout the period of the All-forcing ensemble integrations. In agreement with the modeling results of Lin et al.
(2015), warming SSTs (cf. figure 1a) impact the shallow branch more strongly than the deep branch. As midlatitude planetary
waves are less important for the shallow branch than for the deep branch (Plumb, 2002; Ueyama et al., 2013), it is reasonable to
expect that a reduction in their strength has a relatively smaller impact on the shallow branch. Volcanic eruptions have a weaker
effect in the lower stratosphere as compared to the middle stratosphere (compare the difference between the blue and green
curves for the years 1963/1964, 1983/1984 and 1991/1992 between Figure 4a and Figure 4d). A possible explanation is that
longwave and near-IR heating due to volcanic aerosols occurs at the level of the aerosols (not shown), and in our integrations
the aerosols are quickly lofted higher in the stratosphere (compare the volcanic influence on temperature as a function of time
in the various levels of  Aquila et al., 2016). This allows for stronger and longer-lasting changes mainly in the "deep" branch of



the BDC. As for volcanoes, changing ODS concentrations impacts the deep branch of the BDC more strongly, and this effect is consistent with the idealized modeling results of Gerber (2012). The colder vortex that follows ozone depletion creates a waveguide higher into the stratosphere, raising the breaking level of Rossby waves and deepening the BDC. Hence, it is to be expected that ozone depletion and recovery has a disproportionate impact on the deep branch. Overall, the aging since in the deep branch since 1990 does not extend to the shallow branch because the two factors that led to aging (Pinatubo and declining ODS concentrations) preferentially impact the deep branch, while the two factors that led to freshening (GHG increases and SST warming) preferentially impact the shallow branch.

In the SH lower stratosphere (Figure 4f), ODS concentrations are the dominant forcing, but the gradual decline in ODS concentrations is balanced out by rising GHG concentrations and mean age is flat since 1990 in the blue All-forcing curve. It is known that ozone-depletion induced polar cooling can directly modulate extratropical wave propagation down to the troposphere (Chen and Held, 2007; Oman et al., 2009; Garfinkel et al., 2013), though future work is needed in order to understand how this influence led to an accelerated shallow branch.

### 3.2.3 Residual Circulation

The same forcings led to structural changes in the tropical residual vertical velocity ($\overline{w*}$, Figure 6). In the all-forcing experiment, tropical upwelling accelerated until 1990 in both the mid-stratosphere and lower stratosphere (i.e. blue line rises in all panels of Figure 6), but has since decreased at 50hPa ( Figure 6a) and higher (not shown). The recent deceleration of $\overline{w*}$ at 50hPa comes about due to competition between changing SSTs and declining ODS concentrations: changing SSTs lead to continual acceleration (i.e. the red curve continues to rise), but ODS recovery leads to a slight deceleration (i.e. the gap between the red curve and green curve gradually decreases after 2000). Because of the eruption of Pinatubo, the deceleration trends starts shortly after 1990 in the all forcing experiment (blue curve) rather than in the late 1990s when ODS concentrations began to decrease (Figure 1a). At 100hPa (Figure 6b), on the other hand, the dominant forcing is SSTs, and $\overline{w*}$ continues to increase throughout the experiment. Similar results are found if we consider the total upwelling mass-flux between the turnaround latitudes (not shown). The implications of these changes for ozone and temperature in the lower stratosphere are discussed in Polvani et al. (2016). Overall, the same forcings that control the age of mid-stratospheric air also control the residual circulation, and these forcings can explain the slowdown of the deep branch of the BDC in the NH since 1990.

### 3.2.4 Summary of Key Forcing Agents

In summary, from 1960 to the late 1980s (the first half of the experiments), ozone depletion, rising GHG, and warming SSTs all led to a decrease in mean age in all regions of the stratosphere. Over the second half of the experiments (since the late 1980s), rising GHGs continue to lead to decreasing mean age, though the decrease is more prominent in the lower stratosphere. However, declining ODS concentrations and the proximity of the start-date to the eruption of Pinatubo lead to an aging trend that is most prominent in the mid-stratosphere. The degree of compensation between these forcings is region-specific, and for the NH-midlatitude mid-stratosphere the volcanic effects and declining ODS concentrations dominate while in the lower stratosphere the SSTs and GHGs dominate. Hence, structural changes occurred in the BDC in our simulations.





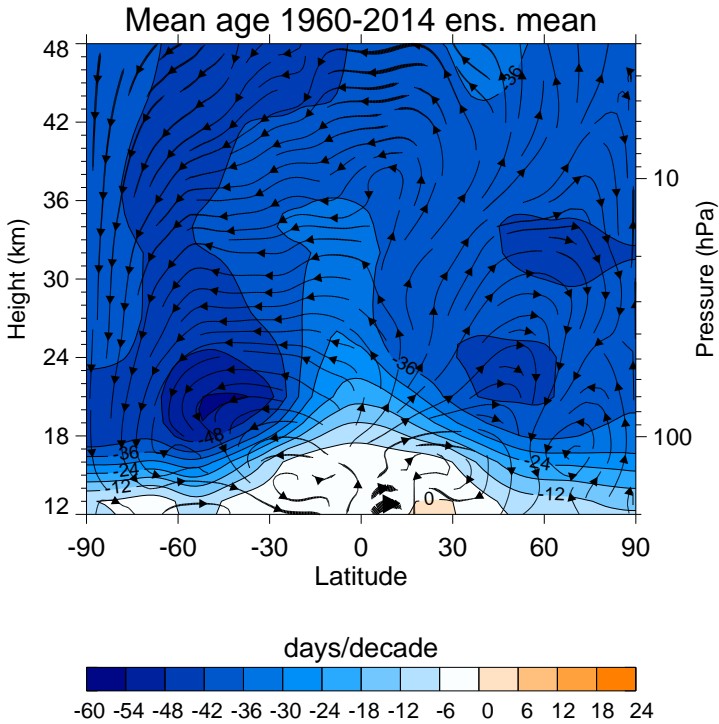

**Figure 2.** Trends in annual averaged BDC from 1960 to 2014 in the ensemble mean of the three all-forcing integrations. Mean age trends are indicated by contours with a contour interval of 6days/decade, and the residual circulation trends are indicated with streamlines. The thickness of the streamline is proportional to the magnitude of the wind speed.





**Figure 3.** (left) As in Figure 2 but for trends in mean age from 1988 to 2014 in the three ensemble members. Note that (a) corresponds to the ensemble member with decreasing mean air and (e) to the ensemble member with aging air in the NH mid-latitude stratosphere. The right column shows the changes in total wave forcing of the BDC (EP flux divergence plus gravity wave drag).



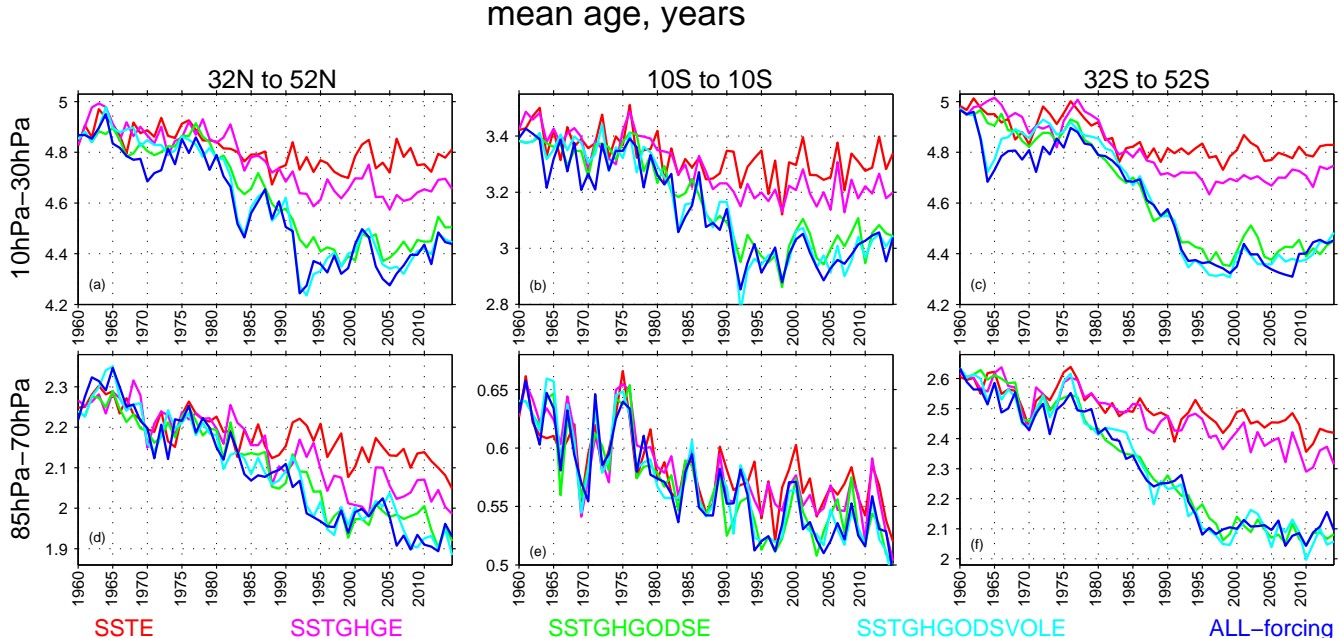

**Figure 4.** Time series of annual averaged mean age in the NH (left), tropics (middle), and SH (right). Each line is for one ensemble mean. Blue is all forcing, red is SST only, and intermediate colors span the other three experiments performed. Changes in the mid-stratosphere (deep branch) are shown in (a)-(c), and changes in the lower stratosphere (shallow branch) are shown in (d)-(f).

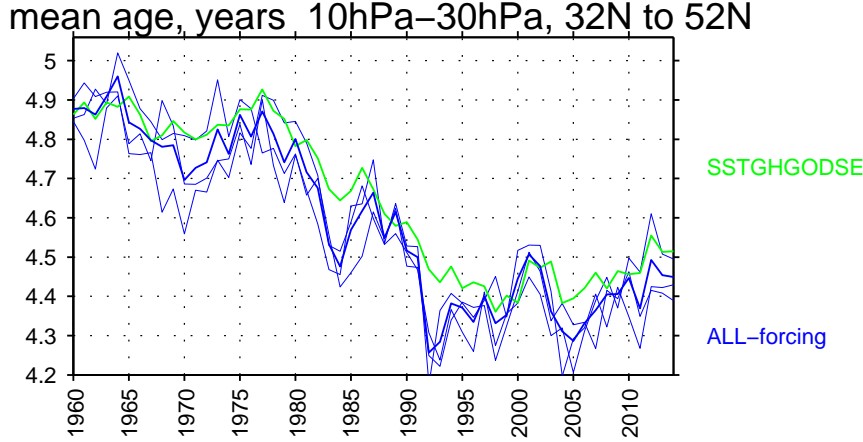

**Figure 5.** Modeled annual averaged mean age in the NH mid-stratosphere between 30hPa and 10hPa in each of the three All-forcing GEOSCCM integrations. Thin lines denote individual integrations, while thick lines denote ensemble means. For clarity, we also include the ensemble mean of the SST+GHG+ODS simulation.





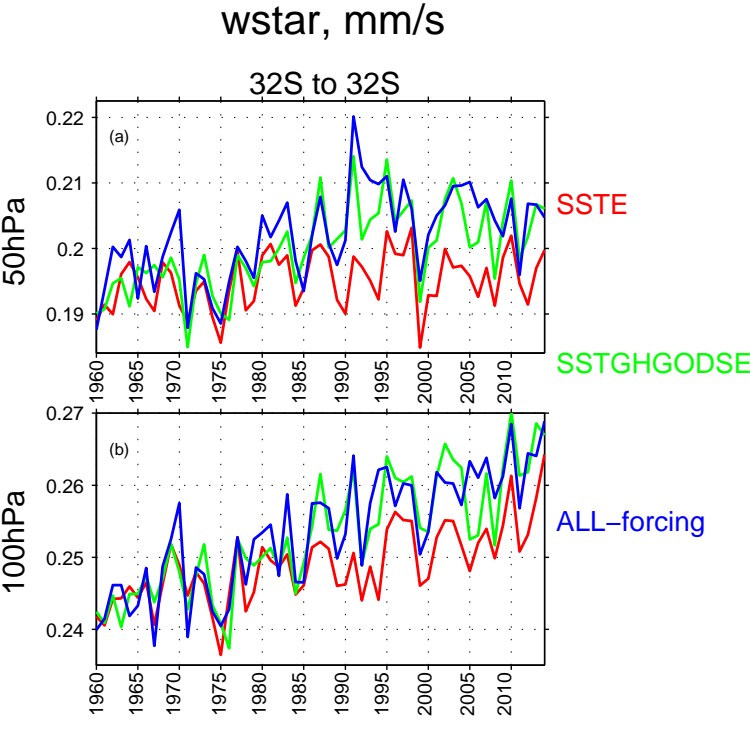

**Figure 6.** Time series of annual averaged tropical $\overline{w*}$ at 50hPa (top) and 100hPa(bottom). Each line is for one ensemble mean. Blue is all forcing, red is SST only, and green is for SSTGHGODS. For clarity, we suppress two of the intermediate experiments.



## 4   Discussion of Observed Changes

It is impossible to directly measure changes in the BDC. However, its evolution can be deduced from trace gas measurements or from reanalyses output data, and here we consider whether the modeled evolution of the BDC in GEOSCCM is consistent with these constraints.

### 4.1   Comparison with BDC Changes inferred from in-situ $CO_2$ and $SF_6$ concentrations since 1975

Balloon measurements of $CO_2$ and $SF_6$ concentrations are available from 1975, and this data does not provide evidence for an acceleration trend in the mid-stratosphere Northern Hemisphere (NH), where mean age actually appears to have increased (Engel et al., 2009; Ray et al., 2014). In particular, the mean age evolution in the figures of Engel et al. (2009) and Ray et al. (2014) indicates pronounced aging since the late 1980s, with earlier changes less clear. As discussed in 3.1, a similar evolution is present in our simulations. In order to make the comparison more precise, we sub-sample the simulated mean age at the latitude and day of each flight analyzed by Ray et al. (2014) and compare it to the mean age reported in figure 7 of Ray et al. (2014). We show all three GEOSCCM members in order to estimate the uncertainty (i.e. internal variability) in the model simulated mean age as deduced from the model's mean age tracer. See Figure 7.

GEOSCCM captures the mean age averaged over this period accurately: the difference between the observations and model for these datapoints is three months. A similar three month offset is evident when comparing GEOSCCM to the mean ages reported by Engel et al. (2009), which falls within the 6-month uncertainty in the observations (Engel et al., 2009). The value of mean age in other regions also agrees well with satellite-based estimates presented in Stiller et al. (2008).

GEOSCCM mean age lies within the error bar for most measurements, and thus is generally consistent with observations. While the weak (non-significant) aging trend noted in observations since 1975 is not present in GEOSCCM, observed and modeled trends agree within the 95% uncertainty level. Note that if we use the wider uncertainties reported by Engel et al. (2009), mean age in GEOSCCM agrees with all balloon flights and trends agree within the 90% uncertainty level. That being said, there is apparently less subseasonal and QBO variability in GEOSCCM than in the observations (and also the tropical leaky pipe model of Ray et al., 2014), and the recent trend towards older air is weaker in GEOSCCM than in the observations. For future work, we will consider whether GEOSCCM is consistent with more recent tracer measurements.

### 4.2   Comparison with BDC Changes since 2002

While extreme caution must be exercised in interpreting a trend over such a short period, we now assess whether BDC changes since 2002 in GEOSCCM are consistent with observational constraints.

Vertically and latitudinally resolved changes in satellite measured $SF_6$ are available since 2002, and Haenel et al. (2015) infer mean age trends from this data (their figure 6). They find that mean age declines in the tropical lower and mid stratosphere south of the equator, and increases in the NH mid-latitudes and in the SH polar stratosphere. We show changes in annual averaged mean age from January 2002 to December 2011 in Figure 8. The model simulates younger mean age in the lower stratosphere in all three ensemble members, but changes higher in the stratosphere are not robust among the various ensemble





members. All statistically significant trends inferred from the satellite data are captured qualitatively by at least one of the three model integrations. (It should not be expected that any single experiment should capture the precise observed trend, as the wave forcing of the BDC differs in any realization of the atmospheric state).

Bönisch et al. (2011) infer BDC changes from $N_2O$ and ozone concentrations and find younger mean age in the NH lower stratosphere, in agreement with our model simulations. Reanalysis-based estimates of mean age over a largely overlapping period also indicate younger mean age in this region (Ploeger et al., 2015). Hence, the NH lower stratosphere recent aging trend simulated in GEOSCCM is consistent with observational constraints.

Mahieu et al. (2014) deduce changes in the BDC from changes in HCl from 2005 to 2010, and they infer an aging of NH (but not SH) stratospheric air over this period. In the ensemble mean in our experiments, SH and NH lower stratospheric mean age decreases (blue line in Figure 4df), though in individual ensemble members aging is present in the NH poleward of 40N (not shown). In the mid-stratosphere, air ages over this period in the NH by over 0.15 years in the ensemble mean (blue line in Figure 4a), and this aging is robust among all three ensemble members; in the SH the changes vary among the ensemble members.

Aschmann et al. (2014) deduce changes in the BDC from changes in ozone, and they infer a lack of acceleration in tropical $\overline{w*}$ above 70hPa since 2002 and an acceleration below 70hPa. Qualitatively similar behavior is evident in Figure 6 - upwelling increases at 100hPa (and more weakly at 85hPa, not shown), but decreases at 70hPa and at all levels higher in the tropical stratosphere. Aschmann et al. (2014) further speculate that this slowdown of the upwelling above 70hPa is associated with the La Nina-like sea surface temperature trends over this period. This conjecture is supported by our modeling results: in the SST-only experiment, $\overline{w*}$ at 100hPa continues to increase over this period, but is largely flat at 50hPa (and also at 70hPa and 30hPa, not shown) . Garfinkel et al. (2015) also noted that recent changes in SSTs (including the La Nina-like sea surface temperature trends) lead to less planetary wave heat flux entering the stratosphere in midlatitudes, and changes in midlatitude planetary waves will impact the deep branch more strongly.

## 4.3 Comparison with BDC Trends Inferred from Reanalysis Output

While deducing trends in the BDC from reanalyses is fraught with danger, we assess whether the modeled evolution of the BDC in GEOSCCM is consistent with available constraints. Abalos et al. (2015) find accelerated tropical $\overline{w*}$ since 1979, while Diallo et al. (2012); Monge-Sanz et al. (2013); Ploeger et al. (2015) find aging of NH air since 1990 in the mid- and upper-stratosphere in trajectory models driven by reanalysis winds. While these reanalysis-forced trajectory studies disagree about the sign of mean age changes in other regions of the stratosphere, they all indicate aging of the NH midlatitude mid-stratosphere since 1990. This apparent inconsistency can be explained by the time period analyzed: our modeling results indicate that trends in the BDC since 1979 and since 1990 differ qualitatively, with only the former indicating stratospheric-wide acceleration of the BDC. Hence, the evolution of the BDC in GEOSCCM and in reanalysis data appears to be consistent.

We find that Pinatubo leads to younger mean age throughout the stratosphere and enhanced tropical upwelling. Figure 2 of Garcia et al. (2011) suggests that similar behavior is present in WACCM, and similar behavior is evident in the mid-stratosphere in SOCOL (Muthers et al., 2016, future work is needed in order to better understand how the details of the prescribed volcanic



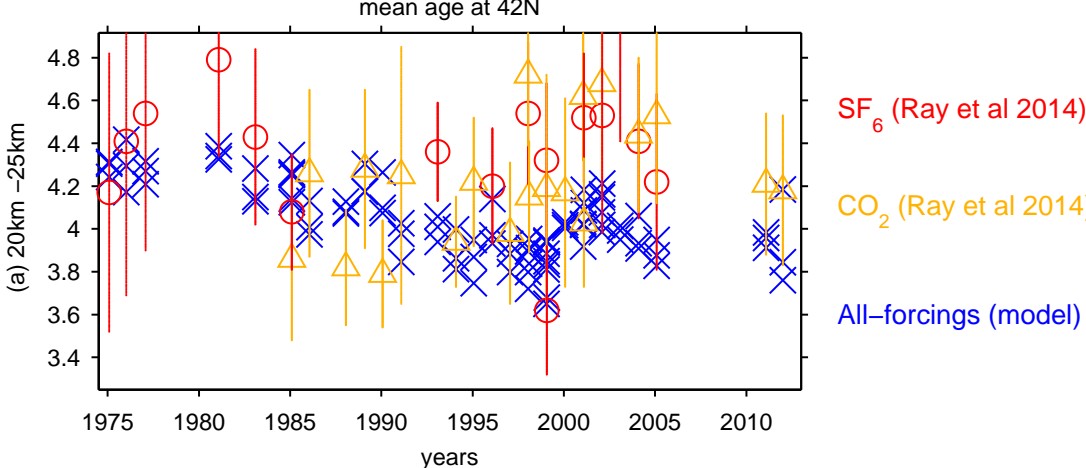

**Figure 7.** Mean age estimates in the data from figure 7 of Ray et al. (2014) in the NH mid-stratosphere between 20km and 25km and in the three all forcing integrations for the same days.

forcing influence the lower stratospheric response). In contrast, Diallo et al. (2012) infer older mean age following Pinatubo using ERA-interim data. However, ERA-interim does not assimilate aerosol data in the radiative scheme (Dee et al., 2011), and thus the strong aerosol loading associated with the volcanic emissions does not affect the tropical heating rates and $\overline{w*}$ in a physically consistent manner. Furthermore, changes in the residual vertical velocity following Pinatubo differ among reanalysis

product and for varying methodologies used for computing the residual vertical velocity (Abalos et al., 2015), and hence the actual response of the BDC to Pinatubo cannot be constrained by reanalysis data. Future work is needed in order to better constrain the response of the BDC to volcanic eruptions using observations.

### 4.4 Summary of the Comparison of GEOSCCM to Observations

In conclusion, the evolution of the BDC in GEOSCCM is generally consistent with observational constraints. There is a

transition between declining mean age throughout the stratosphere before the late 1980s and regionally-specific changes in mean age afterwards (including aging in the mid-latitude mid-stratosphere in the NH). The statement that is often made that climate models simulate a decreasing age throughout the stratosphere only applies over long time periods, and is not the case for the past 25 years when we have most tracer measurements.

### 5 Conclusions

The Brewer-Dobson Circulation (BDC) and its changes have important implications for both stratospheric and tropospheric climate as well as stratospheric ozone chemistry (SPARC-CCMVal, 2010; World Meteorological Organization, 2011, 2014;





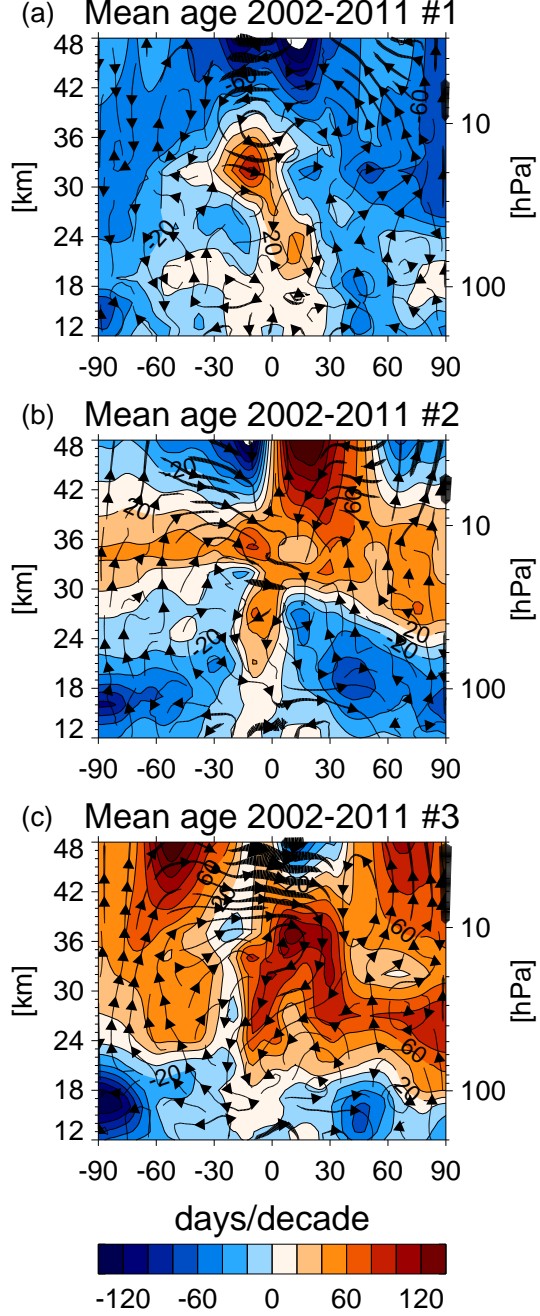

**Figure 8.** As in Figure 2 but for changes in the BDC from January 2002 to December 2011 in the annual mean of the three all-forcing ensembles. Note that the contour interval differs from Figure 2.





Manzini et al., 2014). Hence, it is crucial to understand 1) the structure of historical changes in the BDC, and 2) the factors that lead to these changes. It is also important, for predicting future changes, to know how well models can simulate historical changes of the BDC as given by available observational constraints.

Analysis of a series of chemistry-climate model experiments of the period January 1960 through December 2014 yielded the following conclusions:

1. Over the full duration of the experiments (i.e for a start-date in 1960), we recover the result from previous modeling studies: anthropogenic climate change leads to acceleration of the BDC throughout the stratosphere. Ozone depletion, rising GHG concentrations, and warming SSTs all led to declining mean age in all regions of the stratosphere.

2. Since the late 1980s, structural changes occurred in the BDC. The BDC accelerated in the lower stratosphere in the NH and tropics, but not in the mid-stratosphere. Specifically, in the mid-stratosphere of the midlatitude NH, age of air has increased by 0.2 years since 1992 and the residual circulation has slowed down, and this aging trend is statistically significant. Hence, models can simulate trends generally consistent with available observations.

3. The source of this structural change is the time varying evolution of the forcing factors. While warming SSTs and rising greenhouse gas concentrations both lead to acceleration of the BDC (consistent with previous work), their influence is stronger in the lower stratosphere. In contrast, volcanic eruptions and ODS concentrations generally impact the deep branch more strongly. Declining ODS concentrations and the proximity of the start of declining ODS concentrations to the eruption of Pinatubo leads to an aging trend since the early 1990s in the midlatitude NH mid-stratosphere. While declining ODS concentrations clearly impact the BDC in these integrations, we acknowledge the caveat that observed ozone recovery is unambiguously statistically significant only in the tropical and mid-latitude upper stratosphere (World Meteorological Organization, 2014). If internal atmospheric variability is taken into consideration, then the start-date of an aging trend in the midlatitude NH mid-stratosphere can be pushed back to the late 1988s.

In light of these results, we wish to emphasize that if one wishes to capture observed historical changes in the BDC, careful attention must be paid to the start and end dates used for trend calculation and the forcings included in a model simulation.

Many questions as to the historical changes in the BDC are left unanswered by this study. Diagnostic output necessary to compute the full age spectrum was not saved for these model experiments and hence we are limited in our ability to quantify mixing changes, but it is conceivable that mixing changes contributed to recent observed mean age trends (Ray et al., 2014; Ploeger et al., 2015). In one ensemble member we found that mean age increases in the NH mid-latitude stratosphere since 1988, but there is some hint that aging trends could begin even earlier if the model evolution in different ensemble members was (unrealistically) spliced. This suggests that additional ensemble members may indicate even stronger aging trends even closer to available observations. Finally, for future work, we plan to more quantitatively compare the model mean age to recent in-situ observations.

*Author contributions.* ???????/





*Acknowledgements.* The work of CIG was supported by the Israel Science Foundation (grant number 1558/14) and by a European Research Council starting grant under the European Union's Horizon 2020 research and innovation programme (grant agreement No 677756). The work of DWW is supported, in part, by grants of the US National Science Foundation to Johns Hopkins University. VAA and LDO thank the NASA MAP program for their support. We also thank Eric Ray for providing data from Figures 7 and 8 of Ray et al. (2014) and for help

5   in interpreting balloon data and its uncertainties. We also thank those involved in model development at GSFC-GMAO. High-performance computing resources were provided by the NASA Center for Climate Simulation (NCCS). Correspondence and requests for data should be addressed to C.I.G. (email: chaim.garfinkel@mail.huji.ac.il).



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





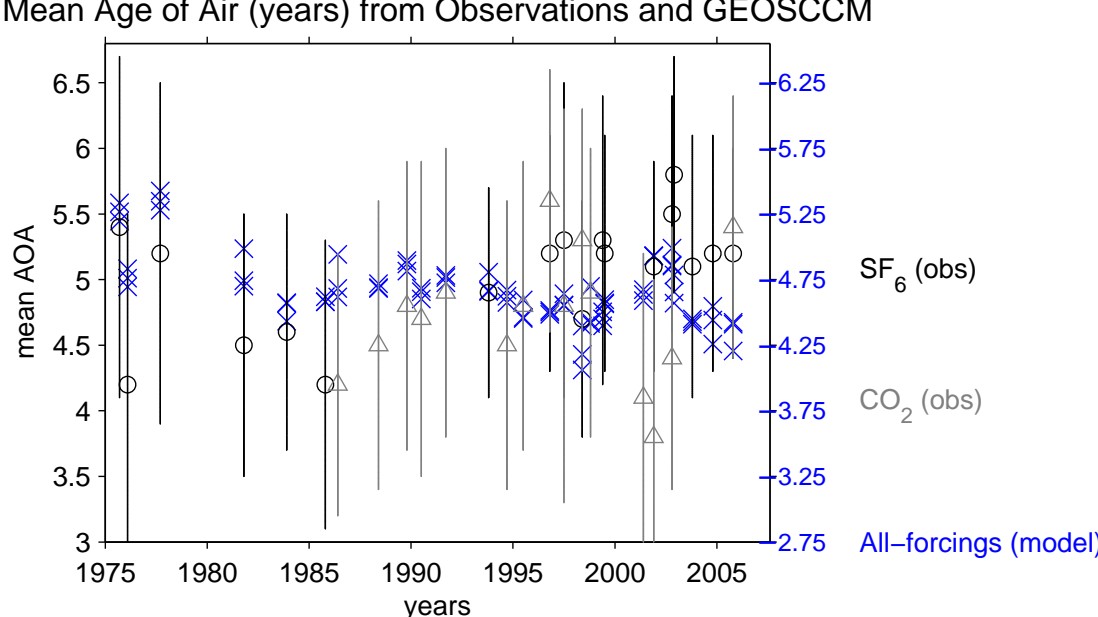

**Figure 9.** mean age estimates in the data from Engel et al. (2009) in the NH mid-stratosphere between 30hPa and 5hPa and in the three all forcing integrations for the same locations and months (results are similar if we use 30hPa to 10hPa). The uncertainty for the observational estimates is taken from Engel et al. (2009), and the uncertainty of the model simulated mean age can be deduced from the intra-ensemble spread. The GEOSCCM mean age is offset by 3 months, i.e. the bias in the mean age (which is less than the 6 month uncertainty in the observed mean age as quoted by Engel et al., 2009).

World Meteorological Organization: Scientific Assessment of Ozone Depletion: 2014, Global Ozone Research and Monitoring Project Rep. No. 55, 2014.