# Peer review of "Time varying changes in the simulated structure of the Brewer Dobson Circulation"

_Atmospheric Chemistry and Physics, 2016_

## Referee Comment (RC1) · Anonymous Referee #4 · 15 Jul 2016

Review of "Time varying changes in the simulated structure of the Brewer Dobson circulation"

By C. Garfinkel et al.

**Recommendation**: accept after minor revision

This is a useful paper that shows that inferences about changes in the BDC derived from a state of the art numerical model are consistent with recent estimates base on observations of trace species.   Further, the study is able to attribute changes in the BDC to various factors (SST, GHG, ODS, volcanoes) by comparing simulations that include one or more of these forcing factors.   A few minor suggestions for changes and clarifications are detailed below.

**Specific Comments** (by page and line number):

(1, 20) "BDC has been deduced from … average time for air parcel…":    The BDC is not deduced from AoA (that is, AoA does not measure the vector circulation $(v^*, w^*)$; instead, it is a proxy for the strength of the BDC, which, furthermore, needs careful interpretation.

(1, 22) "differences": It is not clear how one establishes "differences" between a scalar field (AoA) and a vector field (the BDC). See previous comment.

(2, 14) "pronounced aging": I think this over-states the findings. For example, Engel et al.'s trend estimate is not significantly different from zero.

(2, 17) "aging of the NH": It might be better to write "increasing age of air in the NH". Certainly, the mid-stratosphere of the NH is not getting older (except insofar as the Earth and all of us upon it are getting older.)

(3, 11) "aging of the mid-latitude NH": Better: "aging of air in the mid-latitude NH".

(3, 20) "GEOSCCM": Does the model reproduce the QBO?    Timing between the QBO and the seasonal cycle can introduce substantial low-frequency, stochastic variability in AoA. This is not relevant to long-term climate change but should be noted, especially if it is present in the model, since low-frequency variability could be misinterpreted as a trend in short records.

(4, 21) "interannual and decadal variability in SST": Are you saying that you used the

smoothed version of SST in your simulations? This is not clear; and it is not a trivial point, as such stochastic, low-frequency variability will add "noise" to the time series and make it difficult to say much about trends over short periods of time (25 years or less, in my experience.)

(7, 26) "statistically significant": In general, it would be useful to quote the 2-sigma values every time a trend in AoA is quoted. That way one can get a quick idea of the 95% significance of any trends mentioned. I understand why you may not want to clutter the contour plots by, for example, shading significant regions, but it is easy enough in the text to quote a trend number ± 2-sd.

(7,28) simulate → simulates

(10, 2) "follows ozone depletion": It is plausible that ozone is responsible for BDC changes in the SH mid-stratosphere. But what about the NH, where ozone changes are minuscule compared to the SH, but where AoA also flattens out after 1990? (cf. Figs. 4a and 4c). This explanation seems incomplete to me.

(10, 24) "the same forcings": Isn't this trivially true? After all, AoA is a proxy for the strength of the circulation. Perhaps you had something more profound in mind, but I do not know what.

(15, 1) "impossible to directly measure changes": It is not clear what this means. If you mean that $(v^*, w^*)$ (and, therefore, changes in the BDC) cannot be measured, that is correct. But the diabatic BDC can be obtained from the thermodynamic + continuity equations, and given "good enough" data for a "sufficiently long" period, it should also be possible to detect trends in the BDC. I would think this is probably a more precise, and less ambiguous method than looking at trace gases.

(15, 19) "aging trend noted in observations": Garcia et al. (2011) have discussed why AoA trends derived from trace species may be misrepresented, even when the trends are corrected for growth rate, so one has to take these trends with a grain (or two) of salt.

(15, 26) "extreme caution": Trends over 10 years are not very useful. They can be formally computed, but they are more likely to be influenced by stochastic variability than by any real long-term forcing. (In this regard, please clarify whether you used observed SST or smoothed observed SST to drive the model.)

I was going to suggest that you delete this section, but I think it actually serves a

useful purpose in illustrating how these trends can be "all over the place", especially above ~70 hPa.    Even in the shallow branch, the results are not very consistent among the 3 simulations shown in Fig. 8. So, a useful message from the present exercise is that one should not base any conclusions on the long-term behavior of the BDC on 10-year trends.

(16, 24) "fraught with danger": This is a bit too dramatic. I would think that inferring trends in the BDC from AoA trends derived from observations is even more ambiguous—yet we do it all the time!

(16, 32) "Pinatubo": Better: "the eruption of Mt. Pinatubo". The volcano itself would be irrelevant, and unknown to most of us, had it not erupted.

(17, 12) "only applies over long periods": This is a very useful point, which we often lose sight of, and I am happy to see it emphasized and illustrated by the results presented here.

---

## Referee Comment (RC2) · Anonymous Referee #1 · 18 Jul 2016

The paper presents an assessment of the trends in stratospheric circulation in model simulations of GEOSCCM, and using specific forcing simulations attributes the trends separately to GHG, ozone depletion and volcanoes. The main points are 1) the model trends are highly sensitive to the period chosen, in particular they change between before and after the late 1980s and 2) the model shows aging trends in the NH for the latter period, in qualitative agreement with observational estimates, which can be attributed to volcanic emissions and the end of ozone depletion. The paper is clearly written, the analysis is very relevant and well-timed and the results are interesting. Nevertheless, the paper would improve if the authors carefully address the following minor comments.

*General comments:*

[Figure]

1- The choice of the year separating the early and late periods is not clearly explained, and the authors change it throughout the paper somewhat arbitrarily (1988 in Fig. 3, 1992 in other parts). In particular, considering the trends from 1992 makes the starting point right at the peak influence of the Pinatubo eruption (Fig. 5). Since there are no observational trends reported from this year, I do not see the point in taking this year as a reference, besides enhancing the NH mid-stratosphere aging trends.

2- The age of air trends shown in Fig. 3 are quite similar to those reported for the ERA-Interim reanalysis and also consistent with balloon-derived measurements, but only for simulation 3. The other two members give completely different patterns in the mid-stratosphere. Hence, a major point to extract from Fig. 3 is that there is very large inter-member spread, due to the strong interannual variability and the relatively short time series. In his sense, the sentence in the abstract on P1 L5-6 is misleading, as it leads to think that the model is robustly producing positive trends in the NH mid-stratosphere since the late 1980s.

*Specific comments:*

- P1 L 23 – P2 L2: and even there it is not fully true, near the tropopause there is strong isentropic mixing.

- P2 L14-15: "Reanalysis data": this has only been shown for the ERA-Interim reanalysis, not for others. - P6 L1-2: The reanalysis studies are limited in their initial date because they need to run trajectories for 10 years before they output age of air.

- P6 L9-11: Is this similar to averaging the trends of each member?

- P6 L31-32: Are the trends in the lower stratosphere significant?

- P17 L 9-13: The trends derived from tracer observations are positive from 1975, not just from the late 1980s. This difference between the model and the observations should be mentioned around this part.

- P 9 L27: This is directly shown for sub-seasonal variability in Abalos et al. (2014)

JAS.

- P10 L3-4: I do not think the ozone depletion impacts only the deep branch of the circulation. For instance Abalos et al. 2015 JGR (Fig. 15) show an impact of ozone depletion in the lower stratosphere BDC in reanalyses, linked to enhanced EP flux propagation and convergence around 50 hPa in DJF. In fact, I do not see a significant difference between the All-forcing curves in the middle and lower stratosphere (Figs. 4c and 4f).

- Fig. 6: Why not show the same latitudinal range and levels as in Fig. 4 for consistency?

- P16 L2-3: This important point in the context of interpreting model trends and comparing them to observations, and could be further emphasized.

P17 L11: (including aging in the mid-latitude mid-stratosphere in the NH): this is only seen in one out of three members (see general comment 1).

- Fig. 8: None of the simulations present an inter-hemispheric dipole in the trends as the observations suggest (e.g. MIPAS). This should be pointed out.

*Technical corrections:*

- P4 L21: has → have

- P7 L28: simulate → simulates

- P7 L 30: relative to the other two?

- P7 L30-31: this sentence is not clear, please rephrase it.

- P8 L 29: GHG gases → GHG

- P9 L20: recover → decrease

- P10 L4 remove "since"

- P15 L2: It is impossible to directly measure → There are no direct measurements

---

## Referee Comment (RC3) · Anonymous Referee #3 · 3 Aug 2016

The study by Garfinkel et al uses a comprehensive set of hindcast simulations with a chemistry-climate model to assess past trends in the Brewer-Dobson-Circulation. They show that the circulation speeds up over longer time scales (i.e. 1960 to present), in agreement with past studies, but trends differ in different regions when considering only the last few decades (since the late 1980s). In particular, they show that their model is able to simulate positive AoA trends in the NH mid-stratosphere, which would be in agreement with tracer measurements there. They further show how different forcings contribute to the different trends, and emphasize that the flat or positive trends in the mid-stratosphere since ~1990 are due to the timing of volcanic eruptions and the recovery of ozone (declining ODS concentrations).

Overall, the paper presents interesting new results that contribute significantly to our understanding of the evolution of AoA over the past decades. The statements are

supported by the results shown, and the paper is overall well written. However, the paper could be strengthened significantly by clearer and more quantitative description and presentation of the results, as detailed below. I suggest that the authors revise the paper according to the following comments.

General comments:

1. Overall, the paper would benefit from a more quantitative assessment of the results. The large number and design of the experiments is very well suited for the purpose, but it would be great to see more quantitative measures of the trends and contributions of different forcings. In particular, the availability of 3 ensemble members for each experiment is a perfect basis to estimate the robustness of trends, and I would strongly suggest that this is done. Some examples: Fig.3: It is hard to deduce quantitative changes in the residual circulation from this Figure. I would suggest to include e.g. a Figure of the trend in tropical upwelling as a function of height to justify the statements in the text better. Also, while the text mentions at many points that trends are "significant", please indicate regions of significance in the Figures (e.g. in Fig. 3). Fig. 4: The ensemble members of the experiments can be used to assess the influence of internal variability (e.g. as seen in Fig. 5 for the all-forcing experiment). I agree that it might be too messy to add all the ensemble members in Fig. 4, but for example adding a shaded region to indicate the variability for each experiment would allow better to distinguish between forced differences and variability. Also, the trends discussed from Fig. 4 should be assessed more quantitative rather than only "by eye" - for example calculate the trends for each region as function of start year for each ensemble member and for the ensemble mean (see below).

2. In general, the idea to reconcile the current results with previous studies (Section 4) is a good addition to the paper. However, the discussion is mostly qualitative, and I was left puzzled at the end whether the agreement is good or not. I suggest to make the comparison more quantitative (e.g. add the trend as function of start year from the Ray/Engel observations to the Figure suggested above) where possible, and shorten

the other parts.

3. The authors argue that the recovery from the influence of ODS contributes to positive trends in AoA in the NH mid-stratosphere, but in the tropics and in particular in the SH, ODS leads to AoA "being flat" (page 9, line 18-22). As the influence of ODS concentrations on ozone and subsequently on dynamics is far stronger in the SH, I don't understand this result. If anything, I would expect that the effect of declining ODS is apparent first in the SH. Also, the "positive trend" in the NH appears mainly due to rising AoA in the last few years (from ∼2005), and this positive trend is already visible (though weaker) in the SSTE and SSTGHGE experiments, so I would think that this might have to do with anomalous SSTs, as you actually mention in Section 3.2.1. (1. SSTs). So why do you conclude that ODS are important for the positive trends, when SSTs might contribute to at least a flattening of the trend? Furthermore, as mentioned before, this discussion is somewhat qualitative. A suggestion would be to calculate trends for each experiment (and ensemble member, to get a measure of uncertainty due to internal variability) as function of start date (see above), to gain a more quantitative assessment of the contributions of different forcing to the trend.

Specific comments:

Abstract, line 6: I would rather emphasize that the trend in NH mid-stratospheric AoA are flat/positive in the ensemble mean than that trends are positive in "a simulation" (=one ensemble member?), as the latter result is far less compelling.

Abstract, line 9: Please make the statement that changes in AoA and trop. upwelling "are similar" more quantitative (e.g. both have similar relative trends?)

Abstract, line 11: ".. and is not necessarily the case.." (as it is in some regions).

Page 2, line 2: "...increased or remained unchanged" (as Engel report insignificant changes).

Section 2 (Methods): I agree that it is not necessary to specify all details of the model

(simulation) if described elsewhere. However, at least a comment on the resolution of the model and the upper model level would be desirable. Furthermore, are the boundary conditions and design of the simulations as those specified e.g. for CCMVal2 or CCMI?

page 4, line 25: is the version of the model used in this study is no longer supported, or the one used in Oman 2009? Please specify.

page 4 , line 30: "long" = X years ? Please specify.

page 6, line 30: The upwelling trends are hard to detect from Fig. 3. Include e.g. a Figure with upwelling trends as function of height (see above).

page 6, line 32: Please indicate regions of significance in Fig. 3 (see above).

page 7, line 16: The changes in the residual circulation should be consistent with changes in wave driving in a self-consistent free-running model, so it's good to show this, but does not add that much information. It would be interesting to look into the mechanisms of why these changes in the wave flux changes occur, but I understand that this is beyond the scope of the paper. Also, I personally would find it more interesting to see the ensemble mean changes in circulation and wave fluxes for the different experiments rather than the individual ensemble member of the all-forcing simulation - I think more on the mechanisms could be learned from them.

page 7, line 30: The argumentation of "anomalies occurring in the same integration.." is very speculative and not very physical. Of course, you can always construct ensemble members that show all sorts of trends - however, the question is how likely this is. Given a certain internal variability plus the forced variations, the likelihood to gain a certain trend over a certain period can be calculated.

Paragraph 4.1: I have to admit I'm left puzzled at the end of this section whether trends/variability of the current study and Ray/Engel are consistent or not. I suggest to add a more quantitative evaluation (see 2. general comment). Furthermore, can you

please clarify what the difference between the Ray and Engel time series are?

page 16, line 1-3: The reasoning here is a little weak. I would think what you want to say is that trends over one decade are strongly influenced by internal (unforced) variability, so that a free-running model is not designed to reproduce this trend. You can estimate this uncertainty from your ensemble members (e.g. trend is X+/- Y years/dec) and argue whether the observed trend lies within this uncertainty range. This makes more sense than saying that different aspects of trends are "captured by at least one ensemble member...".

page 16, line 5: Which time period is considered in Bönisch (2011), and do trends only agree in sign or also in magnitude?

page 16, line 6: "NH lower stratosphere aging trend": here AoA decreases, right? so "aging trend" is maybe not the correct wording?

Section 4.2: In general, the discussion in this Section is very qualitative, and for me it is hard to follow which observed aspects are in agreement with the model results and which are not. Please clarify.

page 16, line 25: Abalos shows that trends among reanalysis, and in particular among different methods to estimate w* strongly vary. E.g. tropical upwelling estimated from diabatic heating rates in ERA-Interim show a deceleration in the NH (see their Fig. 11, Fig. 14), consistent with Ploeger and Diallo (that use diabatic heating rates from ERA-Interim). Please clarify.

page 17, line 1ff: I think you have to differentiate in which region you look for the response to volcanic eruptions, as the response is likely spatially not homogeneous. Note that while ERA-Interim does not assimilate aerosol data, it does assimilate temperatures and thus might be able to capture the influence of volcanos indirectly.

page 19, line 27 ff: see above (comment on page 16, line 1ff): the likelihood to obtain a certain trend can be calculated when the internal variability is known, which you

could estimate from the ensemble members. And/or the trend (quantitative!) plus the uncertainty can be calculated, and you can argue whether the observed trend lies within the modelled trend uncertainty. This would make for a much stronger statement then speculating about other ensemble members.

Technical:

page 10, line 4: "the aging since...": delete since

Fig. 6: 32S to 32N (replace S by N)
* * *

---

## Referee Comment (RC4) · Anonymous Referee #5 · 15 Aug 2016

**General comment:**

This paper presents GEOSCCM model simulations of the Brewer-Dobson circulation (BDC) and mean age over the past decades. The authors consider different ensemble members from their simulation which show a very different evolution of the BDC. One member even shows increasing mean age in the NH since 1988, similar to observed mean age trends. In the lower stratosphere, on the contrary, the BDC continues to accelerate. Hence, structural changes evolve in the BDC after 1988. ODS and volcanic eruptions are identified as the main forcing agents of these structural changes.

This paper addresses a very timely aspect of stratospheric dynamics and transport and will definitely be very interesting to a large readership. However, I have three major concerns which the authors need to consider before I can recommend publication.

[Figure]

**Major comments:**

1) Model vs. real atmosphere:

As mentioned above, it is very interesting to see how strongly simulated mean age trends (on decadal time scales) depend on internal variability, and therefore considering the different ensemble members is a very good idea. However, to me the main question remains: which ensemble member is closest to reality and hence most reliable? And this question is not addressed in the paper.

Clearly, if started from different initial conditions the system evolves differently. For instance, there will likely be significant differences in the representation of the QBO between the three members. The authors try to discuss dynamical differences based on residual circulation and EP-flux in Sect. 3.1 (Fig. 3) - but in my opinion this discussion needs to be further substantiated. The following two main questions should be answered:

1) What exactly causes the differences in wave flux and residual circulation between the ensemble members (e.g., differences in the background flow)?

2) Which member has dynamical characteristics closest to the real atmosphere (e.g., a comparison of the QBO with observations could be enlightening)?

If the conclusion of the paper should be "models can simulate trends generally consistent with observations" (P19, L12) it needs to be confirmed that also the underlying dynamics is consistent with available observations or reanalysis data (similar statement on P17, L9), such that the resulting age trend is not due to cancelling effects of errors in different modes of variability. Moreover, I think the wording of the above conclusion is too strong: Only one of the considered ensemble members shows some consistency with observations – and further it is not clear why. If the authors want to make the statement that "BDC in GEOSCCM is generally consistent with observational constraints",

both dynamical quantities (as mentioned above) and mean age need to be more thoroughly compared to available observations. Unless these comparisons are done, the main conclusion should be rephrased rather as: "current model uncertainties due to the representation of internal variability are so large that simulations may be consistent with available observations".

In this context, I don't agree with the commentary of Sect. 4.3 that "deducing trends in the BDC from reanalyses is fraught with danger", while "the modeled evolution of the BDC in GEOSCCM is consistent with available constraint" (P16, L24ff). Mean age simulations based on reanalysis data have been extensively compared to available balloon borne and MIPAS satellite mean age observations and are consistent within observational uncertainties (e.g., Diallo et al., 2012; Ploeger et al., 2015). Furthermore, dynamical variability in the reanalyses is consistent with available observational constraints by definition. The 2002-2011 trend of ensemble member 3 shows aging in the NH similar to MIPAS (e.g., Haenel et al., 2015, Fig. 9) and reanalysis driven simulations. However, negative mean age trends in the SH as observed by MIPAS and consistently simulated by reanalysis-driven models are not simulated by GEOSCCM. Overall, I don't agree with the statement that GCMs are better for estimating decadal (!) trends than reanalysis and I recommend removing Sect. 4.3 (see also my specific comment regarding the representation of volcanic effects in reanalysis).

2) Attribution of mean age trends:

Although the authors explain in the beginning that "mean age is an integrated measure of the total transport" and "only in the tropical lower stratosphere can be thought of as dominated by vertical advection" (P1, L24ff), the following analysis aims to directly link mean age variability with residual circulation (e.g., Fig. 3 and Sect. 3.1). It is known that mixing processes have a strong impact on mean age and its trends (e.g., Neu and Plumb, 1999; Garny at al., 2014) - so what about these effects? I think Fig. 3 can be very misleading in relating the mean age trend just to the residual circulation without including the additional mixing effects, and these likely matter. For instance, why is the

residual circulation and wave flux trend in the NH (northward 30N and above 18km) almost the same for ensemble members 1 and 3, but the resulting mean age trends very different (negative vs. positive NH age trends for members 1 and 2)? I think these mixing effects need to be either analyzed or at least a more careful discussion is needed.

**Specific comments:**

P4, L31: What diagnostics are included in the model? If some measure for mixing could be easily added, this would significantly strengthen the analysis presented here (see my Major comment 2).

P8, L20ff: The discussion here is not entirely clear to me: Why is the "no change in mean age" caused by a "reduction in planetary wave flux entering the stratosphere"? Shouldn't this reduction cause a weakening circulation and increasing mean age? Overall, I have the feeling that the paragraph here is more a discussion than belonging to the results section, as it just discusses the presented mean age changes against the background of published literature.

P10, L5: "... factors that led to aging (Pinatubo ...)..." seems not an optimal choice of wording to me. The direct effect of Pinatubo is to decrease mean age (e.g., Fig. 4a). What you mean here is that this decrease of mean age due to Pinatubo causes a stronger aging trend, as Pinatubo is at the beginning of the considered period. Please improve the wording.

P15, L6ff (section 4.1): The sentence "GEOSCCM mean age lies within the error bar for most measurements, and thus is generally consistent with observations" seems too strong to me. Even if GEOSCCM age lies within the error bars of most of the observations, the model clearly underestimates the mean age after 2000. Further, I think Ray et al. (2014) mapped to 42N equivalent (!) latitude. Is your mean age also sampled at equivalent latitude or just latitude (as I read from the text)? Please clarify.

P16, L29: It was shown by Abalos et al. (2015) and Ploeger et al. (2015) that there is NO "apparent inconsistency" between increasing mean age in the NH between 2002-2012 and an acceleration of the residual circulation in the long-term if mixing effects on mean age and the appropriate time period are taken into account. Please clarify what you mean here.

P17, L1: Diallo et al. (2012) showed increasing mean age after volcanic eruptions only at lower levels in the stratosphere around 19km, and this is indeed consistent with the GCM based results of Muthers et al. (2016) (see their Fig. 3). Hence, there is no "contrast" between the two papers - both are very consistent! The authors are right in saying that ERA-Interim does not assimilate aerosol data. However, parts of the volcanic aerosol effect is included in the reanalysis due to assimilating observed temperatures. And can we be sure that the representation of volcanic aerosol in climate models is correct (e.g., amount of injected aerosol, injection height, ...)? Why, for instance, don't we see an effect of Pinatubo in the SH (Fig. 4c/f) although Pinatubo's effect on temperature appears rather symmetric about the equator – even stronger in the SH (see Fujiwara et al., 2015, Fig. 5)? Hence, I again doubt that decadal trends from climate models are more realistic than from reanalysis-based simulations (see also my Major Comment 1).

Figure 1, caption: Include a decription of the smoothed curve in (a) in the caption. In the model simulation the version without smoothing is used, right? Please clarify in the text.

Figure 3: The statistical significance of the trend is only mentioned in the text but not plotted. It would be better to plot the trend and its significance in the same figure (e.g. as additional shading).

Figure 6: It would be helpful to see also the SSTGHG case in the figure, to estimate the ODS-effect. And why not showing the net upwelling mass flux averaged between turn-around latitudes here? This should give a much more reliable measure of net

upwelling than the flux averaged between fixed latitudes (and as far as I understand, the authors have calculated this already).

Figure 7: It would be helpful to include also the linear trend values for both observed and GEOSCCM simulated mean age in the figure (or in the caption).

**Technical corrections:**

P1, L11: I would better say: "...and is not NECESSARILY the case..."

P1, L23: The $*$ for defining the residual circulation vertical velocity is not raised, it should read $\overline{w}^*$. This occurs several times also at later places in the paper.

P4, L20: $CO_2$ should not be in italics.

P4, L29: I would cite Hall and Plumb (1994) or Waugh and Hall (2002) here, for the calculation of mean age from the linearly increasing tracer.

P6, L8: The number of years "n" should be upper case.

P7, L8: "...resolved waveS..."

P7, L28: "... one ensemble member simulateS..."

P8, L7ff Ray et al. (2010) also found that a "moderate increase in the horizontal mixing into the tropics" has to be assumed in their leaky pipe model (this is also related to my Major Comment 2).

P10, L1: "...concentrations impact..."

P10, L4: "...aging in the deep branch..."

P16, L6: "...recent aging trend...' - I think you mean decreasing mean age?

P16, L16: "...but decreases at 70hPa..." I guess you mean 50hPa, right? At least this is the level shown in your Fig. 6.

Figure 1, caption: Blank missing before "total solar irradiance".

Figure 3, caption: "...decreasing mean AGE OF air..."

Figure 6, title: I guess you mean "32S to 32N", and not "32S to 32S"?

Figure 6, caption: Blank missing after "...100hPa"

P15, L13: "...age tracer (see Fig. 7)."

---

## Author Comment (AC1) · 9 Oct 2016

The paper presents an assessment of the trends in stratospheric circulation in model simulations of GEOSCCM, and using specific forcing simulations attributes the trends separately to GHG, ozone depletion and volcanoes. The main points are 1) the model trends are highly sensitive to the period chosen, in particular they change

between before and after the late 1980s and 2) the model shows aging trends in the NH for the latter period, in qualitative agreement with observational estimates, which can be attributed to volcanic emissions and the end of ozone depletion. The paper is clearly written, the analysis is very relevant and well-timed and the results are interesting. Nevertheless, the paper would improve if the authors carefully address the following minor comments.

General comments:
1- The choice of the year separating the early and late periods is not clearly explained, and the authors change it throughout the paper somewhat arbitrarily (1988 in Fig. 3, 1992 in other parts). In particular, considering the trends from 1992 makes the starting point right at the peak influence of the Pinatubo eruption (Fig. 5). Since there are no observational trends reported from this year, I do not see the point in taking this year as a reference, besides enhancing the NH mid-stratosphere aging trends.

**There are two complementary ways of defining the start date for the trends. The first is to blindly adopt the start date used in previous work (motivating 1988 as the start date), and the second is to choose the start date based on the physical mechanisms that underlie the evolution (motivating 1992 as the start date). We believe that both are justifiable.**

**That being said, we did jump back-and-forth between these two in the original version of the manuscript, and never explicitly justified our choice of 1992 as one of the possible start dates. We now explicitly list a start date of 1992 in the methods section, and more consistently use 1992 after we first start using it.**

**The net effect is that in the revised manuscript, we start with 1988 as the start date, but motivated by our results, we transition over to 1992 towards the beginning of the results section.**

**We also include two new figures where we explicitly explore dependence on the start-date for the trend calculation.**

[Figure]

Figure R1: Trend in tropical mass upwelling at 50hPa and 100hPa for each all-forcing ensemble member separately and for the ensemble mean for start-dates ranging from 1960 through 2002. The vertical line corresponds to the 95% uncertainty range for each trend.

[Figure]

mean age trend (months/decade) until 2014 as a function of start-date

All #1    All #2    All #3    All ensemble mean

**Figure R2: Trend in mean age in the NH, tropics, and SH for each all-forcing ensemble member separately and for the ensemble mean for start-dates ranging from 1960 through 2002. The vertical line corresponds to the 95% uncertainty range for each trend. The high uncertainty in panel (c ) arises because of the large reduction in the effective degrees of freedom due to the high autocorrelation of the residuals (cf. Santer et al 2008).**

2.  The age of air trends shown in Fig. 3 are quite similar to those reported for the ERA-Interim reanalysis and also consistent with balloon-derived measurements, but only for simulation 3. The other two members give completely different patterns in the mid-stratosphere. Hence, a major point to extract from Fig. 3 is that there is very large inter-member spread, due to the strong interannual variability and the relatively short time series. In his sense, the sentence in the abstract on P1 L5-6 is misleading, as it leads to think that the model is robustly producing positive trends in the NH mid-stratosphere since the late 1980s.

**We agree that there is very large inter-model spread; the revised version of the paper now emphasizes this point in the conclusion section with the new sentence "In addition, it should not be expected that any single model experiment should capture the precise observed trend or necessarily resemble a second integration using that same model, as the wave forcing of the BDC differs in any realization of the atmospheric state."  We also changed the sentence in the abstract to: "does not accelerate in the ensemble mean, and even decelerates in one ensemble member"**

Specific comments:

-P1 L 23 – P2 L2: and even there it is not fully true, near the tropopause there is strong isentropic mixing.

**We changed this to "There is no reason to expect the two metrics to indicate a similar time-evolution"**

-P2 L14-15: "Reanalysis data": this has only been shown for the ERA-Interimreanalysis, not for others.

**This has been clarified**

- P6 L1-2: The reanalysis studies are limited in their initial date because they need to run trajectories for 10 years before they output age of air.

**This has been clarified**

-P6 L9-11: Is this similar to averaging the trends of each member?

**The mean trend is similar, but the uncertainty differs.**

-P6 L31-32: Are the trends in the lower stratosphere significant?

**Trends at 100hPa and 85hPa between 20degrees and 60degrees are significant in all three ensemble members. The new figure R2 shows this explicitly, and we plan on including this figure in the revised paper.**

-P17 L 9-13: The trends derived from tracer observations are positive from 1975, not just from the late 1980s. This difference between the model and the observations should be mentioned around this part.

**If you look closely at the figures of Engel et al and Ray et al, there is a short-lived freshening trend from the start of the datasets up until the mid-1980s. The trend using the Ray et al data from 1975-2012 is 0.04years/decade while for the shorter period 1992-2012 it is 0.14 years/decade. Calculating the trend starting in 1975 mixes together two very different regimes of the BDC (i.e. an ozone depletion induced acceleration with a slow deceleration).**

-P 9 L27: This is directly shown for sub-seasonal variability in Abalos et al. (2014) JAS.

**A citation has been added.**

-P10 L3-4: I do not think the ozone depletion impacts only the deep branch of the circulation. For instance Abalos et al. 2015 JGR (Fig. 15) show an impact of ozone depletion in the lower stratosphere BDC in reanalyses, linked to enhanced EP flux propagation and convergence around 50 hPa in DJF. In fact, I do not see a significant difference between the All-forcing curves in the middle and lower stratosphere (Figs. 4c and 4f).

**Yes, the reviewer is correct. We noted in the next paragraph that for the SH lower branch, ozone depletion matters, but we agree the language here was imprecise. This has been corrected**

-Fig. 6: Why not show the same latitudinal range and levels as in Fig. 4 for consistency?

**We pick 32S to 32 N because of its proximity to the turnaround latitudes, however results are similar though much noisier if we focus on 10S-10N like for figure 4 (see below). We show 50hPa and 100hPa as these are below the pressure levels we show for mean age. We show 100hPa instead of 85hPa as the acceleration of w* is more pronounced at 100hPa than at 85hPa (see below).**

[Figure]
**wstar, mm/s**

[Figure]

**Figure R3: Evolution of annual averaged w\* in the tropics at 100hPa and 85hPa for a range of different definitions of the tropics.**

**In any event, in the revised version of the paper, we will show the evolution as the total mass flux between the turnaround latitudes (see figure R4). We also have added a new figure showing the trend at a range of pressure levels (see figure R5a below).**

[Figure]

**Figure R4: Annual averaged upward tropical mass from in the SST-only, SSTGHGODS, and All forcing ensemble means at 50Pa and 100hPa.**

[Figure]

**Figure R5a: trend in tropical upwelling mass flux in the lower stratosphere for the full duration of the experiments and for the period after 1992, for the ensemble mean of each experiment.**

-P16 L2-3: This important point in the context of interpreting model trends and comparing them to observations, and could be further emphasized.

**We agree, the parenthesis have been removed and we now repeat this point in the conclusions section**

P17 L11: (including aging in the mid-latitude mid-stratosphere in the NH): this is only seen in one out of three members (see general comment 1).

**We have changed to "including the possibility of aging"**

- Fig. 8: None of the simulations present an inter-hemispheric dipole in the trends as the observations suggest (e.g. MIPAS). This should be pointed out.

**We now point out this difference with MIPAS explicitly**

Technical corrections:

-P4 L21: has ! have

**corrected**

-P7 L28: simulate ! simulates

**corrected**

-P7 L 30: relative to the other two?

**Yes, relative to the other two, corrected**

-P7 L30-31: this sentence is not clear, please rephrase it.

**We removed this sentence**

-P8 L 29: GHG gases ! GHG

**corrected**

-P9 L20: recover ! decrease

**corrected**

-P10 L4 remove "since"

**corrected**

- P15 L2: It is impossible to directly measure ! There are no direct measurements

**corrected**

The study by Garfinkel et al uses a comprehensive set of hindcast simulations with a chemistry-climate model to assess past trends in the Brewer-Dobson-Circulation. They show that the circulation speeds up over longer time scales (i.e. 1960 to present), in agreement with past studies, but trends differ in different regions when considering only the last few decades (since the late 1980s). In particular, they show that their model is able to simulate positive AoA trends in the NH mid-stratosphere, which would be in agreement with tracer measurements there. They further show how different forcings contribute to the different trends, and emphasize that the flat or positive trends in the mid-stratosphere since 1990 are due to the

timing of volcanic eruptions and the recovery of ozone (declining ODS concentrations).

Overall, the paper presents interesting new results that contribute significantly to our understanding of the evolution of AoA over the past decades. The statements are supported by the results shown, and the paper is overall well written. However, the paper could be strengthened significantly by clearer and more quantitative description and presentation of the results, as detailed below. I suggest that the authors revise the paper according to the following comments.

General comments:

1.Overall, the paper would benefit from a more quantitative assessment of the results. The large number and design of the experiments is very well suited for the purpose, but it would be great to see more quantitative measures of the trends and contributions of different forcings. In particular, the availability of 3 ensemble members for each experiment is a perfect basis to estimate the robustness of trends, and I would strongly suggest that this is done.

Some examples: Fig.3: It is hard to deduce quantitative changes in the residual circulation from this Figure. I would suggest to include e.g. a Figure of the trend in tropical upwelling as a function of height to justify the statements in the text better. Also, while the text mentions at many points that trends are "significant", please indicate regions of significance in the Figures (e.g. in Fig. 3). Fig. 4: The ensemble members of the experiments can be used to assess the influence of internal variability (e.g. as seen in Fig. 5 for the all-forcing experiment). I agree that it might be too messy to add all the ensemble members in Fig. 4, but for example adding a shaded region to indicate the variability for each experiment would allow better to distinguish between forced differences and variability. Also, the trends discussed from Fig. 4 should be assessed more quantitative rather than only "by eye" - for example calculate the trends for each region as function of start year for each ensemble member and for the ensemble mean (see below).

**We agree that we need to do a better job of conveying statistical significance. Note that our attempts at including significance information on the existing figures would lead to unreadable figures, and hence we will include four new figures in the revised manuscript. Figures R1 and R2 shown above will be included in the revised manuscript. These figures explicitly follow your last suggestion.**

**Figure R5a above follows another of your suggestions, and we plan to include it as well in the revised manuscript. It shows the trend in upwelling as a function of height to better justify the statements in the text. Figure R5b is similar to R5a, but for mean**

age in the NH mid-latitudes. It too helps quantify aspects in the text that previously were discussed mostly qualitatively.

[Figure]

Figure R5b: trend in tropical upwelling mass flux in the lower stratosphere for the full duration of the experiments and for the period after 1992, for the ensemble mean of each experiment.

Finally, we made figures showing the range of trends for all three integrations for all five experiments, but as the reviewer intuited there is quite a lot of internal variability especially for trends over the past few decades. See below for an example. We believe this figure isn't particularly helpful, and decided not to include it.

[Figure]

**Figure R6: trend in mean age in the mid stratosphere in the NH for the full duration of the experiments and for the period after 1992, for each experiment. We do not intend to include this figure in the revised manuscript, as internal variability swamps out the forced signal for 23 year trends. This point is already made in Figures R1 and R2, so this figure is redundant.**

2.In general, the idea to reconcile the current results with previous studies (Section 4) is a good addition to the paper. However, the discussion is mostly qualitative, and I was left puzzled at the end whether the agreement is good or not. I suggest to make the comparison more quantitative (e.g. add the trend as function of start year from the Ray/Engel observations to the Figure suggested above) where possible, and shorten the other parts.

**In order to quantify the comparison to Engel/Ray, we would need 3D daily mean age. Unfortunately, this data wasn't saved (only daily lat vs pressure and 3D monthly mean). We deliberated whether to include this section at all in the paper given these data limitations, and decided to still include it. Namely, we don't expect the differences between 3D daily when mapped to equivalent latitude (as done by Ray et al) to differ too drastically from a conventional zonal average at least in midlatitudes in summer (the season for most of these flights). Given these limitations, we are somewhat reluctant to be too quantitative.**

**That being said, we now include the trend and the uncertainties for the period 1992 to the end of the balloon flights for all three ensemble members and for the two papers directly on the figure. One of the ensemble members agrees quite closely with the balloon measurements (0.12±0.10 years/decade for one of the ensemble members, and 0.14±0.14 years/decade for the balloon measurements).**

[Figure]

**We have also significantly shortened sections 4.2 and 4.3, as suggested.**

3. The authors argue that the recovery from the influence of ODS contributes to positive trends in AoA in the NH mid-stratosphere, but in the tropics and in particular in the SH, ODS leads to AoA "being flat" (page 9, line 18-22). As the influence of ODS concentrations on ozone and subsequently on dynamics is far stronger in the SH, I don't understand this result. If anything, I would expect that the effect of declining ODS is apparent first in the SH. Also, the "positive trend" in the NH appears mainly due to rising AoA in the last few years (from 2005), and this positive trend is already visible (though weaker) in the SSTE and SSTGHGE experiments, so I would think that this might have to do with anomalous SSTs, as you actually mention in Section 3.2.1. (1. SSTs). So why do you conclude that ODS are important for the positive trends, when SSTs might contribute to at least a flattening of the trend? Furthermore, as mentioned before, this discussion is somewhat qualitative. A suggestion would be to calculate trends for each experiment (and ensemble member, to get a measure of uncertainty due to internal variability) as function of start date (see above), to gain a more quantitative assessment of the contributions of different forcing to the trend.

**It is worth noting that in the mid-stratosphere in the tropics and SH, there is a trend toward older air too. See figures R1 and R2. Specifically, there isn't a difference in mean age trends between the NH and tropics. In terms of the SH, there are two factors leading to non-significant trends in mean age. First, the correction of Santer et al 2008 for the autocorrelation of residuals leads to huge error bars. Second, the eruption of Pinatubo appears to have impacted the NH more strongly than the SH. Namely, the gap between the green and cyan curves is far larger in figure 4a than in figure 4b and 4c. Mt. Pinatubo is located at 15N and most of the aerosols stay in the NH in our experiments, so the impact on mean age in the NH is stronger than in the**

tropics or SH. This is now noted. That being said, we also note that SSTs alone lead to aging in the NH mid-stratosphere.

Figures R1, R2, R5a, and R5b, which will be included in the revised manuscript, will give a more quantitative assessment of the contribution of different forcings to the trend. As discussed above, we elect not to show each ensemble member for the 4 experiments that are more idealized than all-forcing, as the internal variability is quite large.

Finally, the last paragraph of section 3.2.1 was confusing and misleading, and it has been heavily revised:

"The same forcings that led to a statistically significant aging trend in the NH since 1992 in the ensemble mean and in two integrations also lead to similar significant aging in the tropics (blue curve in Figure 3b and 6b). Specifically, changes in this region are dominated by ODS concentrations, and since ODS concentrations decrease after the late-1990s (Figure 1c), mean age increases despite rising GHG concentrations. As for the NH, the earliest start-date of significant aging trends is in the early 1990s due to the eruption of Mt. Pinatubo. ODS concentrations also dominate the SH mid-stratospheric mean age evolution (Figure 6c), and since ODS concentrations decrease after the late 1990s, mean age is flat since 1992 (i.e. not significantly different from no trend) in the all-forcing ensemble as well. Note that the influence of the eruption of Mt. Pinatubo is weaker in the SH than in the NH; Mt. Pinatubo is located at 15N, and in our experiments the majority of the aerosols stay in the NH."

Specific comments:

Abstract, line 6: I would rather emphasize that the trend in NH mid-stratospheric AoA are flat/positive in the ensemble mean than that trends are positive in "a simulation" (=one ensemble member?), as the latter result is far less compelling.

We changed this to does not accelerate in the ensemble mean, and even decelerates in one ensemble member

Abstract, line 9: Please make the statement that changes in AoA and trop. upwelling "are similar" more quantitative (e.g. both have similar relative trends?)

**We now clarify that both indicate a lack of recent acceleration**

Abstract, line 11: ".. and is not necessarily the case.." (as it is in some regions).

**"Necessarily" has been inserted**

Page 2, line 2: "...increased or remained unchanged" (as Engel report insignificant changes).

**"remain unchanged" has been inserted**

Section 2 (Methods): I agree that it is not necessary to specify all details of the model (simulation) if described elsewhere. However, at least a comment on the resolution of the model and the upper model level would be desirable. Furthermore, are the boundary conditions and design of the simulations as those specified e.g. for CCMVal2 or CCMI?

**We now specify the resolution, upper boundary, and number of layers (2x2.5, 0.01hPa, and 72 levels). The boundary conditions used for GHG and ODS are clarified in the paper as well – we use the standard CCMI and RCP4.5 forcings).**

page 4, line 25: is the version of the model used in this study is no longer supported, or the one used in Oman 2009? Please specify.

**The model version used for this paper is no longer supported. The NASA supercomputer facilities no longer support the older intel fortran compiler.**

page 4 , line 30: "long" = X years ? Please specify.

**10 years, added**

page 6, line 30: The upwelling trends are hard to detect from Fig. 3. Include e.g. a Figure with upwelling trends as function of height (see above).

**Yes, we agree. Such a figure will be included (see figure R5a)**

page 6, line 32: Please indicate regions of significance in Fig. 3 (see above).

**All of our attempts to add significance directly on the figure led to a very noisy and unreadable figure. Instead of this, we show significance for select regions as a function of start date for the trend in figure R1 and R2, which will be included in the revised paper.**

page 7, line 16: The changes in the residual circulation should be consistent with changes in wave driving in a self-consistent free-running model, so it's good to show this, but does not add that much information. It would be interesting to look into the mechanisms of why these changes in the wave flux changes occur, but I understand that this is beyond the scope of the paper. Also, I personally would find it more interesting to see the ensemble mean changes in circulation and wave fluxes for the different experiments rather than the individual ensemble member of the all-forcing simulation - I think more on the mechanisms could be learned from them.

**We have looked at differences in the circulation response among the various integrations, but even the circulation response is tightly coupled to the wave fluxes so in the end we didn't learn much. The difference appears to just be unforced differences in wave fluxes. In a new supplemental section (see response to reviewer 5) we now address changes in the circulation associated with these changes in wave fluxes.**

**We have looked at changes in circulation and wave fluxes in each experiment individually, though the results are indeed beyond the scope of this paper. Aquila et al (2016) describes changes in temperature for each experiment.**

page 7, line 30: The argumentation of "anomalies occurring in the same integration.." is very speculative and not very physical. Of course, you can always construct ensemble members that show all sorts of trends - however, the question is how likely this is. Given a certain internal variability plus the forced variations, the likelihood to gain a certain trend over a certain period can be calculated.

**We removed this sentence and the associated argument.**

Paragraph 4.1: I have to admit I'm left puzzled at the end of this section whether trends/variability of the current study and Ray/Engel are consistent or not. I suggest

to add a more quantitative evaluation (see 2. general comment). Furthermore, can you please clarify what the difference between the Ray and Engel time series are?

**As discussed above, we are unable to process the data in exactly the same way that Engel and Ray did, and hence we can't directly compare apples to apples. However, over the period 1992 to 2012, the trend in one of the ensemble members agrees quite closely with that from Ray et al. See the revised figure below.**

**We now clarify in the introduction that Ray et al argues that the large uncertainty estimates on the trends presented by Engel et al are overly conservative, and that this NH mid-stratosphere aging trend is statistically significant.**

[Figure]

page 16, line 1-3: The reasoning here is a little weak. I would think what you want to say is that trends over one decade are strongly influenced by internal (unforced) variability, so that a free-running model is not designed to reproduce this trend. You can estimate this uncertainty from your ensemble members (e.g. trend is X+/- Y years/dec) and argue whether the observed trend lies within this uncertainty range. This makes more sense than saying that different aspects of trends are "captured by at least one ensemble member...".

**We have removed the remark about at least one of the ensemble members capturing qualitatively the observed trends, and now focus discussion in this paragraph on whether the model can capture the inter-hemispheric dipole.**

page 16, line 5: Which time period is considered in Bönisch (2011), and do trends only agree in sign or also in magnitude?

**They consider changes since 1980 in their paper, but mainly focus on changes that occurred after 2000. The decrease they find is about 1 month/decade over the full period since 1980 which agrees quantitatively with that in our model. However, we decided to remove discussion of the Bonisch paper from this section entirely.**

page 16, line 6: "NH lower stratosphere aging trend": here AoA decreases, right? so "aging trend" is maybe not the correct wording?

**Corrected**

Section 4.2: In general, the discussion in this Section is very qualitative, and for me it is hard to follow which observed aspects are in agreement with the model results and which are not. Please clarify.

**We have significantly shortened section 4.2, as including a detailed comparison to all previous studies that focus on the short period since 2000 is really a distraction from the main points of our paper. We have left in the figure showing that trends among the three ensemble members aren't robust, which is sufficient to convey that trends over such a short period are due more to internal variability than to any climate forcing. We now discuss this figure in more detail.**

**More generally, we believe that there is room for a community paper that quantifies exactly where models and observations disagree and agree, but our paper is not the right location for such a comparison with observations only taken over so short a period. In the final paragraph of the paper, we highlight that we (as a community) need to do a better job of comparing changes in the BDC as inferred from trace gases to those simulated by models.**

page 16, line 25: Abalos shows that trends among reanalysis, and in particular among different methods to estimate w* strongly vary. E.g. tropical upwelling estimated from diabatic heating rates in ERA-Interim show a deceleration in the NH (see their Fig. 11, Fig. 14), consistent with Ploeger and Diallo (that use diabatic heating rates from ERA-Interim). Please clarify.

**Yes, the reviewer is correct. However, note that Monge-Sanz et al 2012 use the divergence of the horizontal wind and still find aging in the NH mid-stratosphere. In any event, we have removed this paragraph from the paper.**

page 17, line 1ff: I think you have to differentiate in which region you look for the response to volcanic eruptions, as the response is likely spatially not homogeneous. Note that while ERA-Interim does not assimilate aerosol data, it does assimilate temperatures and thus might be able to capture the influence of volcanos indirectly.

Indeed we agree that the response is likely not spatially homogeneous. We didn't emphasize this point before, but Pinatubo appears to have impacted the NH mid-stratosphere more strongly than other regions (at least in our experiments). We now mention this.

The issue with assimilating temperature and not aerosol burden (and thus radiative heating) directly is that the causality in the thermodynamic equation might be reversed. An increase in radiative heating in the stratosphere (as occurred after Pinatubo) can either lead to faster upwelling or to warmer temperatures in order to balance the thermodynamic equation. In GEOSCCM, it leads to both (the temperature response in GEOSCCM is shown in Aquila et al 2016).

In contrast, in a reanalysis product which only assimilates the warmer temperatures but doesn't know that diabatic heating has changed, one could imagine that the balance in the thermodynamic equation will be restored by decreasing w* (i.e. that the warmer temperatures are associated with anomalous descent and thus adiabatic warming). The reanalysis isn't informed that the cause of the warming is an anomaly in diabatic heating, and thus tries to maintain balance incorrectly. This is shown clearly by Abalos et al 2015.

Perhaps more crucially, Diallo et al 2012 use the diabatic heating rate methodology to define w*. Abalos et al 2015 show that depending on which method is used to define w*, the response to Pinatubo is different. In particular, the diabatic heating rate methodology shows deceleration of w* associated with diabatic cooling. In GEOSCCM, there is anomalous diabatic heating after the eruption of Pinatubo (due to anomalous shortwave absorption). We have added this to the text.

The discussion of the effects of the eruption of Pinatubo now reads:

"We find that the eruption of Mt. Pinatubo leads to younger mean age throughout the stratosphere and enhanced tropical upwelling. Figure 2 of Garcia et al 2011 suggests that similar behavior is present in WACCM. Similar behavior is evident in the mid-stratosphere in SOCOL, though not in the lower stratosphere. Diallo et al 2012 also

infer older mean age in the lower stratosphere following Pinatubo using ERA-interim data. However, changes in the residual vertical velocity following Pinatubo differ among reanalysis product and for varying methodologies used for computing the residual vertical velocity (Abalos et al 2015), and hence the actual response of the BDC to Pinatubo cannot be constrained by existing reanalysis data. Specifically, Diallo et al 2012 use diabatic heating rates in ERA-interim to define w*, and tropical diabatic heating rates show cooling after Pinatubo in the reanalyses (figure 1 of Abalos et al 2015) but warm in GEOSCCM due to increased shortwave heating from the aerosal plume (not shown). Future work is needed in order to better constrain the response of the BDC to volcanic eruptions using observations."

It is worth noting that the response to the eruption of Mt. Pinatubo is stronger in the NH mid-stratosphere as compared to the SH. The likely cause of this is as follows: Mt. Pinatubo is located at 15N, and in our experiments the majority of the aerosols stay in the NH. Therefore, the increased shortwave heating is stronger in the NH. Future work is needed in order to explore sensitivity to the details of the prescribed volcanic forcing in the model. "

page 19, line 27 ff: see above (comment on page 16, line 1ff): the likelihood to obtain a certain trend can be calculated when the internal variability is known, which you could estimate from the ensemble members. And/or the trend (quantitative!) plus the uncertainty can be calculated, and you can argue whether the observed trend lies within the modelled trend uncertainty. This would make for a much stronger statement then speculating about other ensemble members.

We removed this sentence

Technical:

page 10, line 4: "the aging since...": delete since

corrected

Fig. 6: 32S to 32N (replace S by N)

corrected

Referee 4

Review of "Time varying changes in the simulated structure of the Brewer Dobson circulation"

By C. Garfinkel et al.
**Recommendation**: accept after minor revision

This is a useful paper that shows that inferences about changes in the BDC derived from a state of the art numerical model are consistent with recent estimates based on observations of trace species. Further, the study is able to attribute changes in the BDC to various factors (SST, GHG, ODS, volcanoes) by comparing simulations that include one or more of these forcing factors. A few minor suggestions for changes and clarifications are detailed below.

**Specific Comments** (by page and line number):
(1, 20) "BDC has been deduced from … average time for air parcel…": The BDC is not deduced from AoA (that is, AoA does not measure the vector circulation $(v^*,w^*)$; instead, it is a proxy for the strength of the BDC, which, furthermore, needs careful interpretation.

**The vector circulation $(v^*,w^*)$ only represents the advective part of the BDC, and does not represent the total transport of mass in the stratosphere. This framework misses mixing, a very important element of the BDC. We therefore very intentionally do not wish to define the BDC as the vector circulation $(v^*,w^*)$. These issues are discussed in the Butchart 2014 review paper. We opt to keep the language as is.**

(1, 22) "differences": It is not clear how one establishes "differences" between a scalar field (AoA) and a vector field (the BDC). See previous comment.

**See comments above.**

(2, 14) "pronounced aging": I think this over-states the findings. For example, Engel et al.'s trend estimate is not significantly different from zero.

**it is worth noting that the trend in Ray et al is significantly different from zero. Nevertheless, we have removed "pronounced".**

(2, 17) "aging of the NH": It might be better to write "increasing age of air in the NH". Certainly, the mid-stratosphere of the NH is not getting older (except insofar as the Earth and all of us upon it are getting older.)

**changed**

(3, 11) "aging of the mid-latitude NH": Better: "aging of air in the mid-latitude NH". **changed**

(3, 20) "GEOSCCM": Does the model reproduce the QBO? Timing between the QBO and the seasonal cycle can introduce substantial low-frequency, stochastic variability in AoA. This is not relevant to long-term climate change but should be noted, especially if it is present in the model, since low-frequency variability could be misinterpreted as a trend in short records.

**The model does spontaneously generate a QBO, though the QBO phase is randomized among the three ensemble members. This has now been noted in the text:**

**"Note that this model version includes a QBO, and that the QBO phase differs among the three ensemble members."**

(4, 21) "interannual and decadal variability in SST": Are you saying that you used the smoothed version of SST in your simulations? This is not clear; and it is not a trivial point, as such stochastic, low-frequency variability will add "noise" to the time series and make it difficult to say much about trends over short periods of time (25 years or less, in my experience.)

**We used the full SST fields. We include the smoothed version in the figure solely to highlight that the SST only experiment includes part of the climate change signal, but also includes e.g. ENSO variability. This has now been clarified:**
**"all experiments were forced with the full time evolving SST fields".**

(7, 26) "statistically significant": In general, it would be useful to quote the 2-sigmavalues every time a trend in AoA is quoted. That way one can get a quick idea of the 95% significance of any trends mentioned. I understand why you may not want to clutter the contour plots by, for example, shading significant regions, but it is easy enough in the text to quote a trend number ± 2-sd.

**Figure R2, which we will include in the revised submission, will quantify the trends and their associated uncertainties. We prefer to present this information in the figures, and not in the text, in order to maintain readability of the text as much as possible.**

(7,28) simulate →simulates
**corrected**

(10, 2) "follows ozone depletion": It is plausible that ozone is responsible for BDC changes in the SH mid-stratosphere. But what about the NH, where ozone changes are minuscule compared to the SH, but where AoA also flattens out after 1990? (cf. Figs. 4a and 4c). This explanation seems incomplete to me.

**If ozone depletion changes upwelling anywhere in the tropics, it can affect age of air in the NH, via mixing. This has now been clarified:**
**"An ozone-induced acceleration of the BDC in the SH and tropics will affect mean age in the NH due to mixing."**

(10, 24) "the same forcings": Isn't this trivially true? After all, AoA is a proxy for the strength of the circulation. Perhaps you had something more profound in mind, but I do not know what.

**It isn't trivially true. One could imagine an exotic change in mixing that affects transport and hence mean age, but has no impact on the residual circulation. (Such a change has been proposed in order to explain part of the historical changes in the BDC.) We prefer to leave the text as is. Given that we don't explicitly diagnose mixing in these simulations, however, we prefer not to explicitly discuss the importance of mixing changes except in a qualitative sense.**

(15, 1) "impossible to directly measure changes": It is not clear what this means. If you mean that (v*, w*) (and, therefore, changes in the BDC) cannot be measured, that is correct. But the diabatic BDC can be obtained from the thermodynamic + continuity equations, and given "good enough" data for a "sufficiently long" period, it should also be possible to detect trends in the BDC. I would think this is probably a more precise, and less ambiguous method than looking at trace gases.

**We have changed to "There are no direct measurements of historical changes in the BDC."**

(15, 19) "aging trend noted in observations": Garcia et al. (2011) have discussed why AoA trends derived from trace species may be misrepresented, even when the trends are corrected for growth rate, so one has to take these trends with a grain (or two) of salt.

**The Ray et al 2014 paper adjusts for these nonlinearities in the growth, yet they still find significant aging of air.   We now clarify this point in the introduction.**

(15, 26) "extreme caution": Trends over 10 years are not very useful. They can be formally computed, but they are more likely to be influenced by stochastic variability than by any real long-term forcing. (In this regard, please clarify whether you used observed SST or smoothed observed SST to drive the model.)
I was going to suggest that you delete this section, but I think it actually serves a useful purpose in illustrating how these trends can be "all over the place", especially above ~70 hPa. Even in the shallow branch, the results are not very consistent among the 3 simulations shown in Fig. 8. So, a useful message from the present exercise is that one should not base any conclusions on the long-term behavior of the BDC on 10-year trends.

**We agree, but for better or worse such short trends are published and interpreted as being meaningful. Indeed the entire point of this section was to point out just how "all over the place" such 10-year trends can be. We have now clarified that the reason for extreme caution is the large stochastic variability in the atmospheric circulation, and that one should not base any conclusions on the long-term behavior of the BDC on 10-year trends.**

(16, 24) "fraught with danger": This is a bit too dramatic. I would think that inferring trends in the BDC from AoA trends derived from observations is even moreambiguous—yet we do it all the time!

**This paragraph has been completely removed from the paper (though we do agree we were a bit too dramatic)**

(16, 32) "Pinatubo": Better: "the eruption of Mt. Pinatubo". The volcano itself would be irrelevant, and unknown to most of us, had it not erupted.
**changed**

(17, 12) "only applies over long periods": This is a very useful point, which we often lose sight of, and I am happy to see it emphasized and illustrated by the results presented here.

**thanks!**

This paper presents GEOSCCM model simulations of the Brewer-Dobson circulation (BDC) and mean age over the past decades. The authors consider different ensemble members from their simulation which show a very different evolution of the BDC. One member even shows increasing mean age in the NH since 1988, similar to observed mean age trends. In the lower stratosphere, on the contrary, the BDC continues to accelerate. Hence, structural changes evolve in the BDC after 1988. ODS and volcanic eruptions are identified as the main forcing agents of these structural changes.

This paper addresses a very timely aspect of stratospheric dynamics and transport and will definitely be very interesting to a large readership. However, I have three major concerns which the authors need to consider before I can recommend publication.

**First of all, we wish to thank the reviewer for their constructive criticism and very close reading of our manuscript. They have led to substantial improvements.**

**Major comments:**

1) Model vs. real atmosphere:

As mentioned above, it is very interesting to see how strongly simulated mean age trends (on decadal time scales) depend on internal variability, and therefore considering the different ensemble members is a very good idea. However, to me the main question remains: which ensemble member is closest to reality and hence most reliable? And this question is not addressed in the paper.

We wish to clarify that these are free-running experiments run with fixed initial and boundary conditions only. It is not particularly informative to ask which ensemble member is most "reliable", as any difference among them is due to unforced

**variability. The motivation for studying historical changes in the BDC in free running climate simulations is *not* to form a best estimate of what actually occurred; for that purpose, reanalyses and/or nudged experiments are far better. Rather, the motivation is two-fold: one, future projections of the BDC can only be produced by free running climate simulations, and these projections are of limited value if the models can't generally capture the past evolution; two, the forcings that caused these historical changes can be systematically diagnosed.**

**That being said, in our response below we do attempt to answer the reviewer's question as to which ensemble member has circulation trends closest to the observed evolution.**

Clearly, if started from different initial conditions the system evolves differently. For instance, there will likely be significant differences in the representation of the QBO between the three members. The authors try to discuss dynamical differences based on residual circulation and EP-flux in Sect. 3.1 (Fig. 3) - but in my opinion this discussion needs to be further substantiated. The following two main questions should be answered:

1)What exactly causes the differences in wave flux and residual circulation between the ensemble members (e.g., differences in the background flow)?

**There is substantial chaotic, stochastic variability in atmospheric eddies. This large amount of stochastic variability is by definition unforced, and has no "cause" other than the chaos inherent in the Navier-Stokes equations. This large amount of stochastic variability underlies the entire premise of ensemble forecasting in operational use all around the globe.**

**While there certainly are differences in the background flow among the ensemble members, this needs to be the case if the wave fluxes differ. Hence, analyzing the background state isn't particularly enlightening (though see below for our attempt). Only on very short (submonthly) timescales can one hope to disentangle wave fluxes from the background state associated with them (e.g. Garfinkel et al 2012; Watson et al 2014 in the context of the QBO).**

2) Which member has dynamical characteristics closest to the real atmosphere (e.g., a comparison of the QBO with observations could be enlightening)? If the conclusion of the paper should be "models can simulate trends generally consistent with observations" (P19, L12) it needs to be confirmed that also the underlying dynamics is consistent with available observations or reanalysis data (similar statement on P17, L9), such that the resulting age trend is not due to cancelling effects of errors in different modes of variability.

**We have removed the words "generally consistent" throughout the paper, as it is vague and not enlightening.**

**The QBO phase in all three is randomized and does not follow observations in any of the three. Please see the figure below (the period before 1997 is cut off for clarity). For any given 5 year period, one ensemble member mimics observations more than the others, but the "best" ensemble member changes with time. In particular, the main difference among the ensemble members in terms of mean age is in the last five years of the experiments, and over these five years the QBO in none of the ensemble members matches that observed. Note that the red line (all #3) is the ensemble member that bears the strongest trend towards a slowdown of the BDC, though its QBO evolution closely resembles a different ensemble member.**

[Figure]

**Figure R7: Tropical zonal wind at 50hPa in the three all-forcing integrations and in the MERRA reanalysis.**

**We now copy and paste from the planned supplementary section, as it follows directly from the reviewers comment**

"In the main body we find that one all-forcing ensemble member simulates an aging trend of mid-stratospheric air since 1988 consistent with observational constraints (labeled ALL #3 in Figures 2 and 5), while a second simulates freshening of air and an accelerated mid-stratospheric BDC (labeled ALL #1 in Figures 2 and 5). It is reasonable to ask which ensemble member is closer to reality as measured by other metrics. It is also reasonable to ask whether the difference in the wave fluxes underlying the different BDC trends leads to differences in temperature and zonal wind trends.  We now explore these issues by comparing the trend in these two ensemble members to that in MERRA and ERAI reanalysis. As the difference in wave forcing between these ensemble members was pronounced mainly in the Southern Hemisphere, we focus on this hemisphere.

Figure R8 compares the temperature trends over the 1988 to 2014 period in each ensemble member to various reanalysis products in order to assess which integration has a trend in the thermal structure closest to observations.

Consistent with the trend towards more wave convergence in ALL #1, there is a warming trends over the pole. In contrast, the trend towards less wave convergence in ALL #3 (which allows the BDC to slow down) is reflected in cooling over the pole. More relevantly, there is an equator-to-pole gradient in the trends: in ALL #3, the pole cools more than the tropics, reflecting a slowdown of the residual circulation. In contrast, in ALL #1 the tropics cools more than the pole reflecting a speedup of the residual circulation.

 These trends can then be compared to those in MERRA and ERAI. First of all, it is worth noting that the two reanalysis products differ as to the magnitude of the trend, though in both there is a tripole pattern in the vertical over the pole, with cooling in the upper and lower stratosphere and warming in the mid-stratosphere. Neither ensemble member captures this tripole pattern. However, both reanalysis products indicate a trend towards cooling upon averaging the trends over the entire SH extratropics, and only ALL #3 captures this feature. As these experiments are all run in climate mode (no nudging), there are differences in the wave fluxes and hence in the temperature trends, so there should not be the expectation that any ensemble member agrees fully with the actual observed trends.

**Figure R9 compares the zonal wind trends over the 1988 to 2014 period in each ensemble member to various reanalysis products. Consistent with the temperature trends, there is an easterly trend in ALL #1 and a westerly trend in ALL #3 in order to maintain thermal wind balance. ALL #3 compares very favorably to the trends in ERAI, though the large differences between MERRA and ERAI preclude any meaningful quantitative comparison.**

**The net effect is that ALL #3 (which simulates a slowdown of the BDC from 1988 to 2014) has circulation trends that more closely resemble reanalysis than ALL #1 (which simulates a speedup of the BDC from 1988 to 2014). However, the correspondence between ALL #3 and reanalysis is not full, but it is not reasonable to expect perfect correspondence as the wave fluxes in any given realization of nature differ. "**

**Figure R8: Trend in annual averaged temperature in the SH from 1988 to 2014 in the (a) MERRA and (b) ERAI reanalyses, and in the all forcing integrations with a trend towards a (c) accelerated BDC and (d) a decelerated BDC.**

[Figure]

[Figure]

**Figure R9: Trend in annual averaged zonal wind in the SH from 1988 to 2014 in the (a) MERRA and (b) ERAI reanalyses, and in the all forcing integrations with a trend towards a (c) accelerated BDC and (d) a decelerated BDC.**

Moreover, I think the wording of the above conclusion is too strong: Only one of the considered ensemble members shows some consistency with observations – and further it is not clear why. If the authors want to make the statement that "BDC in GEOSCCM is generally consistent with observational constraints", both dynamical quantities (as mentioned above) and mean age need to be more thoroughly compared to available observations. Unless these comparisons are done, the main conclusion should be rephrased rather as: "current model uncertainties due to the representation of internal variability are so large that simulations may be consistent with available observations".

**We wish to clarify that internal variability isn't a source of model uncertainty. It is an inherent part of the climate system. Even if only one ensemble member shows an evolution consistent with observational constraints, that tells you that models can capture the observed trends given the large amount of internal variability in**

the climate system, as part of the observed trend may be unforced. We now clarify this when we first introduce the model:

> "This internal variability is not a source of model uncertainty; rather it is an inherent part of the climate system. If the BDC in one ensemble member, but not in the other two, evolves consistently with observational constraints, one can reasonably conclude that models can capture the observed trends if part of the observed trend was due to internal variability and was not forced."

More generally, the motivation for studying historical changes in the BDC in free running climate simulations is two-fold: one, future projections of the BDC can only be produced by free running climate simulations, and these projections are of limited value if the models can't generally capture the past evolution; two, the forcings that caused these historical changes can be systematically diagnosed. In contrast, the motivation is *not* to form a best estimate of what actually occurred; for that cause, reanalyses are far better. We realize we never stated this explicitly in the original submission, but we have now added this when we introduce the model as well:

> "The motivation for studying historical changes in the BDC in free running climate simulations is *not* to form a best estimate of the actual historical evolution; for that purpose, reanalyses and/or nudged experiments are far better. Rather, the motivation is two-fold: one, future projections of the BDC can only be produced by free running climate simulations, and these projections are of limited value if a model's simulation of the past is inconsistent with observational constraints; two, assuming the model is capable of following the observed evolution, the forcings that caused these historical changes can be systematically diagnosed by sequentially adding these forcings. "

In this context, I don't agree with the commentary of Sect. 4.3 that "deducing trends in the BDC from reanalyses is fraught with danger", while "the modeled evolution of the BDC in GEOSCCM is consistent with available constraint" (P16, L24ff).

**Our terminology "fraught with danger" was a bit too much, but in any event as you suggest below we have significantly shortened section 4.3 and removed this paragraph. The "consistent with available constraints" comment is no longer present in this section.**

Mean age simulations based on reanalysis data have been extensively compared to available balloon borne and MIPAS satellite mean age observations and are consistent within observational uncertainties (e.g., Diallo et al., 2012; Ploeger et al., 2015). Furthermore, dynamical variability in the reanalyses is consistent with available observational constraints by definition. The 2002-2011 trend of ensemble member 3 shows aging in the NH similar to MIPAS (e.g., Haenel et al., 2015, Fig. 9) and reanalysis driven simulations. However, negative mean age trends in the SH as observed by MIPAS and consistently simulated by reanalysis-driven models are not simulated by GEOSCCM. Overall, I don't agree with the statement that GCMs are better for estimating decadal (!) trends than reanalysis and I recommend removing Sect. 4.3 (see also my specific comment regarding the representation of volcanic effects in reanalysis).

**We never meant to claim that GCMs are better for estimating decadal trends than reanalysis data - we apologize for the confusing wording in the initial submission. We now discuss explicitly the NH-SH asymmetry in the trends as inferred from MIPAS.**

**"Vertically and latitudinally resolved changes in satellite measured SF_6 are available since 2002, and Haenel et al 2015 infer mean age trends from this data (their figure 6). They find that mean age declines in the tropical lower and mid stratosphere south of the equator, and increases in the NH mid-latitudes and in the SH polar stratosphere. We show changes in annual averaged mean age from January 2002 to December 2011 in Figure 12. The model simulates younger mean age in the lower stratosphere in all three ensemble members, but changes higher in the stratosphere are not robust among the various ensemble members. These intra-ensemble differences highlight the fact that one should not base any conclusions on the long-term behavior of the BDC on 10-year trends, as trends over one decade are strongly influenced by unforced (internal) variability. While not one of the ensemble members capture the inter-hemispheric dipole in the trends as suggested by satellite data (though individual integrations separately capture half of the dipole), we suggest that such a correspondence should not necessarily be expected as the wave forcing of the BDC differs in any realization of the atmospheric state (e.g. see the discussion in Santer et al 2008)."**

**Section 4.3 has now been removed.**

2) Attribution of mean age trends:

Although the authors explain in the beginning that "mean age is an integrated measure of the total transport" and "only in the tropical lower stratosphere can be thought of as dominated by vertical advection" (P1, L24ff), the following analysis aims to directly link mean age variability with residual circulation (e.g., Fig. 3 and Sect. 3.1). It is known that mixing processes have a strong impact on mean age and its trends (e.g., Neu and Plumb, 1999; Garny at al., 2014) - so what about these effects? I think Fig. 3 can be very misleading in relating the mean age trend just to the residual circulation without including the additional mixing effects, and these likely matter. For instance, why is the residual circulation and wave flux trend in the NH (northward 30N and above 18km) almost the same for ensemble members 1 and 3, but the resulting mean age trends very different (negative vs. positive NH age trends for members 1 and 2)? I think these mixing effects need to be either analyzed or at least a more careful discussion is needed.

**Unfortunately, the age spectrum diagnostic was not turned on when these simulations were first performed, and hence we cannot explicitly examine mixing. We therefore are reluctant to conclude anything as to mixing trends.**

**To answer your question regarding figure 3, we have added " Note that a difference in wave fluxes in the SH can influence mean age in the NH because mean age is an integral measure of transport, and thus changes in the BDC in the tropics due to wave flux changes in the SH can impact transport pathway into the NH."**

**Specific comments:**

P4, L31: What diagnostics are included in the model? If some measure for mixing could be easily added, this would significantly strengthen the analysis presented here (see my Major comment 2).

**The age spectrum diagnostic is the best diagnostic for mixing, but it was not turned on when these simulations were first performed. It is not possible to reconstruct this**

**information. w\* and v\* were saved, as was the EP flux divergence and gravity wave zonal torque.**

P8, L20ff: The discussion here is not entirely clear to me: Why is the "no change in mean age" caused by a "reduction in planetary wave flux entering the stratosphere"? Shouldn't this reduction cause a weakening circulation and increasing mean age? Overall, I have the feeling that the paragraph here is more a discussion than belonging to the results section, as it just discusses the presented mean age changes against the background of published literature.

**Recent changes in SSTs have caused a tug-of-air of effects. Widespread warming of SSTs causes an acceleration of the BDC (many mechanisms have been presented as to why this must be), but the specific pattern of changes in SSTs over the past 35 years has led to a decline in planetary wave flux (as noted by Garfinkel et al 2015) which in isolation causes a deceleration. The net effect is that there is no change in the BDC in the SST only simulation. We have rewritten this paragraph to enhance clarity.**

**"The likely cause of this is a tug-of-war of opposing effects. On the one hand, gradual warming of the oceans in isolation leads to an acceleration of the BDC (Oman et al 2009). On the other hand, the spatial pattern of recent changes in SSTs have led to a decline in planetary wave flux (especially wave 1) entering the NH stratosphere in midlatitudes (Garfinkel et al 2015): the vertical component of the Eliassen Palm flux at 100hPa area averaged between 40N-80N declines in all three all forcing and in all three SST only experiments, and the decrease is statistically significant at the 95% level in the ensemble mean in both January through March (the focus of Garfinkel et al 2015) and in the annual mean. This decline in upward propagating midlatitude planetary waves at 100hPa impacts the deep branch more strongly (Plumb 2002, Ueyama et al 2013). Hence, it is not surprising that little change occurred over the last 30 years of the integration."**

**We seriously considered moving the paragraph to the discussion, but elected to keep it here.**

P10, L5: "... factors that led to aging (Pinatubo ...)..." seems not an optimal choice of wording to me. The direct effect of Pinatubo is to decrease mean age (e.g., Fig. 4a). What you mean here is that this decrease of mean age due to Pinatubo causes a stronger aging trend, as Pinatubo is at the beginning of the considered period. Please improve the wording.

**Now reads: "recovery from the Pinatubo eruption"**

P15, L6ff (section 4.1): The sentence "GEOSCCM mean age lies within the error bar for most measurements, and thus is generally consistent with observations" seems too strong to me. Even if GEOSCCM age lies within the error bars of most of the observations, the model clearly underestimates the mean age after 2000. Further, I think Ray et al. (2014) mapped to 42N equivalent (!) latitude. Is your mean age also sampled at equivalent latitude or just latitude (as I read from the text)? Please clarify.

**We acknowledge in the text that "the recent trend towards older air is weaker in GEOSCCM"**

**The reviewer is correct when saying that we calculate mean age at 42N, while Ray et al calculate at equivalent latitude of 42N. In order to transform to equivalent latitude, one needs 3D daily mean age. Unfortunately, this data wasn't saved (only daily zonal mean and 3D monthly mean). We deliberated whether to include this section at all in the paper given these data limitations, and decided to still include it. Namely, we don't expect the differences between 3D daily when mapped to equivalent latitude (as done by Ray et al) to differ too drastically from daily zonally averaged, especially in summer when most of these balloon flights occurred. However, given these limitations, we focus mainly on qualitative aspects of the agreement. We now explicitly note this difference in data processing. This difference notwithstanding, we have added the trends directly to the figure. Over the period of 1992-2012, one of the ensemble members agrees very closely with reanalysis ((0.12±0.10 years/decade for one of the ensemble members, and 0.14±0.14 years/decade for the balloon measurements). See the figure below.**

[Figure]

**We have also significantly shortened sections 4.2 and 4.3, as suggested.**

P16, L29: It was shown by Abalos et al. (2015) and Ploeger et al. (2015) that there is NO "apparent inconsistency" between increasing mean age in the NH between 2002- 2012 and an acceleration of the residual circulation in the long-term if mixing effects on mean age and the appropriate time period are taken into account. Please clarify what you mean here.

**We agree that changes in mixing could explain this difference. In any event, this paragraph has been removed from the paper.**

P17, L1: Diallo et al. (2012) showed increasing mean age after volcanic eruptions only at lower levels in the stratosphere around 19km, and this is indeed consistent with the GCM based results of Muthers et al. (2016) (see their Fig. 3). Hence, there is no "contrast" between the two papers - both are very consistent! The authors are right in saying that ERA-Interim does not assimilate aerosol data. However, parts of the volcanic aerosol effect is included in the reanalysis due to assimilating observed temperatures. And can we be sure that the representation of volcanic aerosol in climate models is correct (e.g., amount of injected aerosol, injection height, ...)? Why, for instance, don't we see an effect of Pinatubo in the SH (Fig. 4c/f) although Pinatubo's effect on temperature appears rather symmetric about the equator – even stronger in the SH (see Fujiwara et al., 2015, Fig. 5)? Hence, I again doubt that decadal trends from climate models are more realistic than from reanalysis-based simulations (see also my Major Comment 1).

**We have modified this paragraph to more clearly note that Muthers et al and Diallo et al agree about the lower stratospheric response. We agree that future work needs to explore the sensitivity of the response to Pinatubo to the details of how the aerosol cloud is inserted; we now state this more clearly as well. Note that the eruption of Pinatubo does affect the SH as well, however the influence is stronger in the NH (the reason for this hemispheric asymmetry is likely because Mt Pinatubo is at 15N, and at least in our experiments most of the aerosols stay in the NH.)**

**That being said, we suggest that the vertical wind product used by Diallo probably isn't ideal if one wants to constrain the response to volcanos. The issue with assimilating temperature and not radiative heating directly is that the causality in the thermodynamic equation might be reversed. An increase in shortwave radiative heating in the stratosphere (as occurred after Pinatubo) can either lead to faster**

upwelling or to warmer temperatures in order to balance the thermodynamic equation. In GEOSCCM, it leads to both (the temperature response in GEOSCCM is shown in Aquila et al 2016 and the upwelling response is shown here and in Aquila et al 2013).

In contrast, in a reanalysis product that only assimilates the warmer temperatures but doesn't know that diabatic heating has changed, one could imagine that the balance in the thermodynamic equation will be restored by decreasing w* (i.e. that the warmer temperatures are associated with anomalous descent and thus adiabatic warming). The reanalysis isn't informed that the cause of the warming is an anomaly in diabatic heating, and thus tries to maintain balance incorrectly. This point is discussed by Abalos et al 2015.

Perhaps more crucially, Diallo et al 2012 use the diabatic heating rate methodology to define w*.  Abalos et al 2015  show that depending on which method is used to define w*, the response to Pinatubo is different. In particular, the diabatic heating rate methodology shows deceleration of w* associated with diabatic cooling. In GEOSCCM, there is anomalous diabatic heating after the eruption of Pinatubo (due to anomalous shortwave absorption). We have added this to the text.

Finally, we agree that un-nudged climate model runs are not as useful for constraining the actual historical changes in the BDC. We never meant to imply this.

The discussion of the effects of the eruption of Pinatubo now reads:

"We find that the eruption of Mt. Pinatubo leads to younger mean age throughout the stratosphere and enhanced tropical upwelling. Figure 2 of Garcia et al 2011 suggests that similar behavior is present in WACCM. Similar behavior is evident in the mid-stratosphere in SOCOL, though not in the lower stratosphere. Diallo et al 2012 also infer older mean age in the lower stratosphere following Pinatubo using ERA-interim data. However,  changes in the residual vertical velocity following Pinatubo  differ among reanalysis product and for varying methodologies used for computing the residual vertical velocity (Abalos et al 2015), and hence the actual response of the BDC to Pinatubo cannot be constrained by existing reanalysis data. Specifically, Diallo et al 2012   use diabatic heating rates in ERA-interim to define w*, and tropical diabatic heating rates  show cooling after Pinatubo  in the reanalyses  (figure 1 of Abalos et al 2015) but warm  in GEOSCCM due to increased shortwave heating from the aerosal plume (not shown). Future work is needed in order to  better constrain the response of the BDC to volcanic eruptions using observations."

**It is worth noting that the response to the eruption of Mt. Pinatubo is stronger in the NH mid-stratosphere as compared to the SH. The likely cause of this is as follows: Mt. Pinatubo is located at 15N, and in our experiments the majority of the aerosols stay in the NH. Therefore, the increased shortwave heating is stronger in the NH. Future work is needed in order to explore sensitivity to the details of the prescribed volcanic forcing in the model. "**

Figure 1, caption: Include a description of the smoothed curve in (a) in the caption. In the model simulation the version without smoothing is used, right? Please clarify in the text.

**The caption has been clarified. The model uses the full SST fields, not the smoothed version. This has also been clarified.**

Figure 3: The statistical significance of the trend is only mentioned in the text but not plotted. It would be better to plot the trend and its significance in the same figure (e.g. as additional shading).

**Our attempts at adding shading led to an unreadable figure. Instead, we have added new figures that show significance as a function of start-date for each ensemble member for selected regions (see figures R1, R2, R5a, and R5b above).**

Figure 6: It would be helpful to see also the SSTGHG case in the figure, to estimate the ODS-effect. And why not showing the net upwelling mass flux averaged between turn-around latitudes here? This should give a much more reliable measure of net upwelling than the flux averaged between fixed latitudes (and as far as I understand, the authors have calculated this already).

**Even with three lines, the figure is already somewhat cluttered. We don't want to add a fourth. Figure R5a, which will be included in the revised manuscript, will allow one to infer the ODS effect. All of the w* figures have been updated to the net upwelling mass flux.**

Figure 7: It would be helpful to include also the linear trend values for both observed and GEOSCCM simulated mean age in the figure (or in the caption).

**This has been done, see the figure above**

**Technical corrections:**

P1, L11: I would better say: "...and is not NECESSARILY the case..."

**Changed**

P1, L23: The for defining the residual circulation vertical velocity is not raised, it should read w . This occurs several times also at later places in the paper.

**We have confirmed that the asterisk in w\* is not raised in our source file.**

P4, L20: $CO_2$ should not be in italics.

**Changed**

P4, L29: I would cite Hall and Plumb (1994) or Waugh and Hall (2002) here, for the calculation of mean age from the linearly increasing tracer.

**Citation added**

P6, L8: The number of years "n" should be upper case.

**Changed**

P7, L8: "...resolved waveS..."

**Changed**

P7, L28: "... one ensemble member simulateS..."

**Changed**

P8, L7ff Ray et al. (2010) also found that a "moderate increase in the horizontal mixing into the tropics" has to be assumed in their leaky pipe model (this is also related to my Major Comment 2).

**We now mention that changes in mixing could be important in section 4, when we revisit observations more directly.**

P10, L1: "...concentrations impact..."

**Changed**

P10, L4: "...aging in the deep branch..."

**Changed**

P16, L6: "...recent aging trend...' - I think you mean decreasing mean age?

**Yes, this has been clarified.**

P16, L16: "...but decreases at 70hPa..." I guess you mean 50hPa, right? At least this is the level shown in your Fig. 6.

**This has now been changed to 50hPa.**

Figure 1, caption: Blank missing before "total solar irradiance".

**Changed**

Figure 3, caption: "...decreasing mean AGE OF air..."

**Changed**

Figure 6, title: I guess you mean "32S to 32N", and not "32S to 32S"? Figure 6, caption: Blank missing after "...100hPa"

**Fixed**

P15, L13: "...age tracer (see Fig. 7)."

**Changed**

---

## Referee Report (RR1)

**Reviewer comment on Garfinkel et al. (2016), ACPD:**

Overall, the authors did a good job in modifying the paper by including new figures and improving the text. I do recommend publication now. However, when reading I still detected a few typoes and places where I have some minor remarks:

**Minor remarks:**

p18/l24: '...the red curve continues to rise in Figure 9a...'
The trend (of the red curve) is indeed positive, but insignificant as Figs. 10bd show. Hence, the wording could be misleading.

p19/l22: '...though the veracity of this assumption should be tested for future work.'
Either your assumption (that in the summertime midlatitude mid-stratosphere equivalent latitude is similar to latitude) is valid, then there is no need to test it further, or it is not - and then you shouldn't use it... I just did a quick look into equivalent latitude calculated from ERA-Interim on 1 August 2013 (12 UTC) at 500K and found that along a latitude of 60°N the equivalent latitude varies between 35°-83°N. Hence, it is not clear to me that your assumption here is a good one... I would suggest to remove the justification of using just latitude (following 'However, ...' in p19/l20, and just state the difference to the analysis by Ray et al.

p19/l24: In my opinion, the 'GEOSCCM captures the mean age averaged over this period accurately' is too strong. There are clear differences between the simulations and the observations by 0.5-1 yr in Fig. 11 (e.g., around year 2000). Further, the 'mean age in other regions also agrees well with satellite-based estimates presented in Stiller et al. (2008)' is purely speculative. If such a strong statement is to be made the comparison should better be shown.

p21/l6: '...and hence the actual response of the BDC to Pinatubo cannot be constrained by existing reanalysis'
Given the good comparison of observed mean age with simulations driven by reanalysis data, in my opinion the wording is too strong. I would suggest to write something like:
'Hence, although simulations driven by ERA-Interim compare well with observed BDC changes it is a challenging question how reliable the BDC response to Pinatubo is represented in current reanalysis data.'

p22/l2: 'mean age has increased by 0.15 years'
Where does the number 0.15 in this rather general statement come from? Fig. 7 (all forcing) in NH shows a trend of 0.05 yr/dec, which would yield an increase of ~0.12 yr over 1992-2014. Fig. 11 presents a trend of 0.12 yr/dec at the locations of Engel-measurements. I'm a bit confused here...

**Technical corrections:**

p2/l13: blank missing after 'unchanged'

p7/l7: '...for mean age in THE selectED region...'

p8/l2: '..can impact THE transport pathway...'

p10/l5: '...in te All-forcing ensemble MEAN, ...'

Figure 7: An explanation of the colors in the figure is missing. I'm sure the same colors are used as in Fig. 6, but they should be also explained in Fig. 7.

p18/l22: no blank after bracket '( Figure 4a...'

p19/l17: '...in SECT. 3.1 ...'

p21/l5: '...among reanalysis productS'

p22/l15: better write '...single FREE-RUNNING model...'

---

## Author Response (AR2)

Overall, the authors did a good job in modifying the paper by including new figures and improving the text. I do recommend publication now. However, when reading I still detected a few typoes and places where I have some minor remarks:

Minor remarks:

p18/l24: '...the red curve continues to rise in Figure 9a...'
The trend (of the red curve) is indeed positive, but insignificant as Figs. 10bd show. Hence, the wording could be misleading.

We have added "though insignificant"

p19/l22: '...though the veracity of this assumption should be tested for future work.'
Either your assumption (that in the summertime midlatitude mid-stratosphere equivalent latitude is similar to latitude) is valid, then there is no need to test it further, or it is not - and then you shouldn't use it... I just did a quick look into equivalent latitude calculated from ERA-Interim on 1 August 2013 (12 UTC) at 500K and found that along a latitude of 60°N the equivalent latitude varies between 35°-83°N. Hence, it is not clear to me that your assumption here is a good one... I would suggest to remove the justification of using just latitude (following 'However, ...' in p19/l20, and just state the difference to the analysis by Ray et al.

We have removed this sentence as suggested, and just state the difference in methodology.

p19/l24: In my opinion, the 'GEOSCCM captures the mean age averaged over this period accurately' is too strong. There are clear differences between the simulations and the observations by 0.5-1 yr in Fig. 11 (e.g., around year 2000). Further, the 'mean age in other regions also agrees well with satellite-based estimates presented in Stiller et al. (2008)' is purely speculative. If such a strong statement is to be made the comparison should better be shown.

We meant it matches the climatological value of mean age – the difference between observations and the model averaged over all of these datapoints is three months. This has now been clarified.
We also clarify that the comparison with Stiller et al 2008 is also referring to the climatological value of mean age.

p21/l6: '...and hence the actual response of the BDC to Pinatubo cannot be constrained by existing reanalysis'
Given the good comparison of observed mean age with simulations driven by reanalysis data, in my opinion the wording is too strong. I would suggest to write something like:
'Hence, although simulations driven by ERA-Interim compare well with observed BDC changes it is a challenging question how reliable the BDC response to Pinatubo is represented in current reanalysis data.'

We have replaced "cannot be constrained with existing reanalysis data" with "is poorly constrained by existing reanalysis data".

If different modern reanalyses products disagree as to the sign of the response to Pinatubo, it is reasonable to argue that reanalyses products offer poor constraints on the response to Pinatubo.

p22/l2: 'mean age has increased by 0.15 years'
Where does the number 0.15 in this rather general statement come from? Fig. 7 (all forcing) in NH shows a trend of 0.05 yr/dec, which would yield an increase of ~0.12 yr over 1992-2014. Fig. 11 presents a trend of 0.12 yr/dec at the locations of Engel-measurements. I'm a bit confused here...

Thank you for pointing this out. We have clarified that this refers to the ensemble mean, and changed this to 0.12 pm 0.09 years per decade. We have also added that the trend in one ensemble member is quantitatively similar to that in available observations.

Technical corrections:

p2/l13: blank missing after 'unchanged'

Thank you for pointing this out. This has been fixed.

p7/l7: '...for mean age in THE selectED region...'

Thank you for pointing this out. This has been fixed.

p8/l2: '..can impact THE transport pathway...'

Thank you for pointing this out. This has been fixed.

p10/l5: '...in te All-forcing ensemble MEAN, ...'

Thank you for pointing this out. This has been fixed.

Figure 7: An explanation of the colors in the figure is missing. I'm sure the same colors are used as in Fig. 6, but they should be also explained in Fig. 7.

Thank you for pointing this out. The figure has been updated.

p18/l22: no blank after bracket '( Figure 4a...'

Thank you for pointing this out. This has been fixed.

p19/l17: '...in SECT. 3.1 ...'

Thank you for pointing this out. This has been fixed.

p21/l5: '...among reanalysis productS'

Thank you for pointing this out. This has been fixed.

p22/l15: better write '...single FREE-RUNNING model...'

Thank you for pointing this out. This has been fixed.